# The Utility of "Even if..." Semifactual Explanation to Optimise Positive Outcomes

**Eoin M. Kenny**[*]
Massachusetts Institute of Technology
Cambridge, MA, U.S.A.
`ekenny@mit.edu`

**Weipeng Huang**[*]
Tencent Security Big Data Lab
Shenzhen, Guangdong Province, China
`fuzzyhuang@tencent.com`

## Abstract

When users receive either a positive or negative outcome from an automated system, Explainable AI (XAI) has almost exclusively focused on how to mutate negative outcomes into positive ones by crossing a decision boundary using counterfactuals (e.g., *"If you earn 2k more, we will accept your loan application"*). Here, we instead focus on *positive* outcomes, and take the novel step of using XAI to optimise them (e.g., *"Even if you wish to half your down-payment, we will still accept your loan application"*). Explanations such as these that employ "even if..." reasoning, and do not cross a decision boundary, are known as semifactuals. To instantiate semifactuals in this context, we introduce the concept of *Gain* (i.e., how much a user stands to benefit from the explanation), and consider the first causal formalisation of semifactuals. Tests on benchmark datasets show our algorithms are better at maximising gain compared to prior work, and that causality is important in the process. Most importantly however, a user study supports our main hypothesis by showing people find semifactual explanations more useful than counterfactuals when they receive the positive outcome of a loan acceptance.

## 1 Introduction

Explainable AI (XAI) is broadly categorised into factual [4, 27, 30] and contrastive explanations [29, 38]. Within contrastive XAI, despite being neglected in comparison to counterfactuals, semifactuals are a major, fundamental part of human explanation, and have long been studied in psychology [9, 35, 49], philosophy [6, 8, 20], and lately computer science [1–3, 29, 34, 52, 62]. They take the form of *"Even if $x$ happened, $y$ would still be the outcome"*. Such reasoning has many potential uses as demonstrated by these prior works, but here we are focused on how semifactuals can help optimise positive outcomes for users, which (to the best of our knowledge) remains completely unexplored.

Our definition of counterfactuals is in line with Wachter et al. [55], where a test instance classified as $c$ must be mutated to cross a decision boundary into class $c'$. Likewise, as established in the literature [1, 29], we define a semifactual as an instance classified as $c$, which must be modified in such a way as to *not* cross a decision boundary (and hence remain class $c$) [29]. In recourse [25, 50], "negative outcomes" (e.g., a loan rejection) are generally mutated to produce "positive outcomes" (e.g., a loan acceptance) for users using counterfactuals. In our setting, we are assuming there was initially a positive outcome, and we are trying to mutate features to produce an even better situation for users, and in doing so *not* cross the boundary into the negative outcome (i.e., using semifactuals).

Historically, counterfactuals have had obvious applications in computer science, such as explaining how to have a bank loan accepted rather than rejected, but applications for semifactuals as less

---

[*]Contributed Equally.
[1]Code available at: `https://github.com/EoinKenny/Semifactual_Recourse_Generation`

37th Conference on Neural Information Processing Systems (NeurIPS 2023).

clear. As such, the usage of semifactuals has often inadvertently defaulted to copying counterfactual research by also explaining negative outcomes (e.g., *"Even if you double your savings, your loan will still be rejected"* [3, 29, 48]). However, such an application for semifactual explanation perhaps has two main issues. Firstly, it is debatable if these explanations convey useful information [1], whilst a counterfactual explaining how to cross a decision boundary and have a loan accepted has obvious utility [24]. Secondly, such explanations make the user's situation seem helpless [35], in that they cannot possibly have their loan accepted, which raises ethical concerns [1]. However, our proposed framework can be used to not only overcome both of these issues, but actively *contribute* to fairness.

Firstly, to try offer useful information for users, we flip the usual recourse problem and consider the user starting from a positive (rather than a negative) outcome. In this setting, consider a user that has had their loan accepted, but might prefer to make a smaller down-payment on a loan application. In this situation, our framework could present an explanation such as *"Even if you half your down-payment, your loan will still be accepted"*, which seems to be more useful than explaining negative outcomes (see Section 6). Secondly, because we are starting from a positive outcome, there is no danger of manipulating people into accepting a negative outcome, which guarantees fairness is this regard. Now, with regards to optimising fairness even further, note that banks are not motivated to share such explanations even though they may help people, because (for example) larger down-payments are associated with lower risk on their behalf [7]. So, the usage of semifactuals in this application has clear potential to actively *encourage* fairness and transparency. As an aside, it is worth noting that although the focus of this paper is on financial applications, this research has broad impact on any domain for which the optimisation of a positive outcome is beneficial. For instance, in medical applications, our framework could present explanations of the form *"Even if you half your dose of drug $x$, you will still be at a low risk for disease $y$"*. This is once again important for optimising fairness because people are frequently over-prescribed medicine with adverse side-effects [47], but due to profit Big Pharma has no incentive to actively encourage this type of transparency. Similar usage of semifactuals have also been proposed in smart agriculture to combat climate change [29].

Our main contributions are: (1) the first explicit exploration of how to optimise positive outcomes with XAI, (2) the problem formulation for this which involved augmenting current semifactual research with the concept of *Gain* (see Section 3.3), and (3) the premiere user test in the XAI literature for semifactuals, showing a clear application in which users find them more useful than counterfactuals.

## 2 Literature Review

When using contrastive explanation to explain loan acceptance decisions, to the best of our knowledge, this has only been explored by McGrath et al. [36]. Specifically, they suggest *positive counterfactuals*, which show "by how much" a user had their loan accepted to help inform them when making future financial decisions. While this is interesting information, we show that users find semifactual explanations more useful in loan acceptance situations than positive counterfactuals (see Section 6).

Semifactual explanation is growing in popularity [3], Kenny & Keane [29] first explored the idea, but focused only on images.[2] Artelt & Hammer [1] used diverse semifactuals to explain why an AI system refuses to make predictions due to having an unacceptably low certainty, but ignore how to explain either positive or negative outcomes. Lu et al. [34] explain spurious patterns with semifactuals using a human-in-the-loop framework in NLP. Zhao et al. [62] proposed a class-to-class variational encoder (C2C-VAR) with low computational cost that can generate semifactual images. Vats et al. [52] used generative models to produce semifactual image explanations for classifications of ulcers. Lastly, for model exploration, Xie et al. [59] sampled semifactual images with a joint Gaussian mixture model, and Dandl et al. [15] proposed deriving semifactual explanations from interpretable region descriptors. In contrast to all these approaches, we are showcasing how semifactuals can be used to optimise positive outcomes for users (notably in causal settings).

From a user perspective, many have discussed the urgent need for comparative tests with semifactuals [29, 37, 40, 48, 56], with Aryal & Keane [3] pointing to the *'paucity of user studies'* in the area. However, the only such tests we are aware of are in the psychological literature over two decades ago [35]. Taking to this challenge, we conduct the first such test directly comparing semifactuals to counterfactuals in the XAI literature (see Section 6).

---

[2]Note there is other work on *a-fortiori* explanations which have similar computational techniques to semifactuals [14, 19, 45], they are justifications of the form *"Because $x$ it true, $y$ must also be true"*.

Our research is related to algorithmic recourse [50] in that we are trying to ensure users are treated fairly by automated systems [24]. In this area, Mothilal et al. [39] explored counterfactual diversity, in that we should be offering users several explanations. In addition, counterfactual robustness has been examined [18], which proposes that generated explanations should be robust to distributional shifts. Lastly, causality has been argued as essential to providing plausible recourse [26]. We see these three facets as being important to our problem setting, and instantiate them in our framework. There are other areas in recourse such as sequential decision making [16, 42], fairness [54], and privacy [43], but we leave their exploration within semifactual explanation for future work.

As an aside, the literature on sufficiency could be conflated with semifactual explanation, as it describes a set of "sufficient" features for a prediction which, in the presence of the other features mutating, mostly doesn't affect the outcome [17, 46, 57]. However the techniques offer no insights for how to generate a meaningful semifactual. More importantly though, if the sufficient features are the only actionable ones, then by definition we can't modify them to create a semifactual.

## 3 Semifactual Framework

In this section, we describe the basic definitions and assumptions for our semifactual framework to optimise positive outcomes for users, before formalising it under the concept of *Gain* (i.e., how much a user stands to benefit from the explanation) in a causal setting, neither of which has been considered before. As an aside, we also show how the established concepts of plausibility, robustness, and diversity can be made fit into the objective to offer better explanations. Finally, we reflect on the theoretical properties of the framework.

### 3.1 Definitions

Let us denote an individual $\mathbf{x} \in \mathcal{X}$ with $k$ *mutable* features $\mathbf{X} = \{X_1, \ldots, X_k\}$. Given the individual $\mathbf{x}$, a set of actions can be applied to $\mathbf{x}$ where each action $a(\mathbf{x})$ is also a $k$-dim vector. As in prior work [25], we apply $a(\mathbf{x})$ and $a$ exchangeably, since the individual $\mathbf{x}$ will always be fixed. We explicitly exclude features that are either *immutable* or *non-actionable*. Adopting Pearl's $do()$ operator [44], an action can be defined as $a(\mathbf{x}) = do(\mathbf{X} := \boldsymbol{\theta})$, or simply $do(\boldsymbol{\theta})$, to force a hard

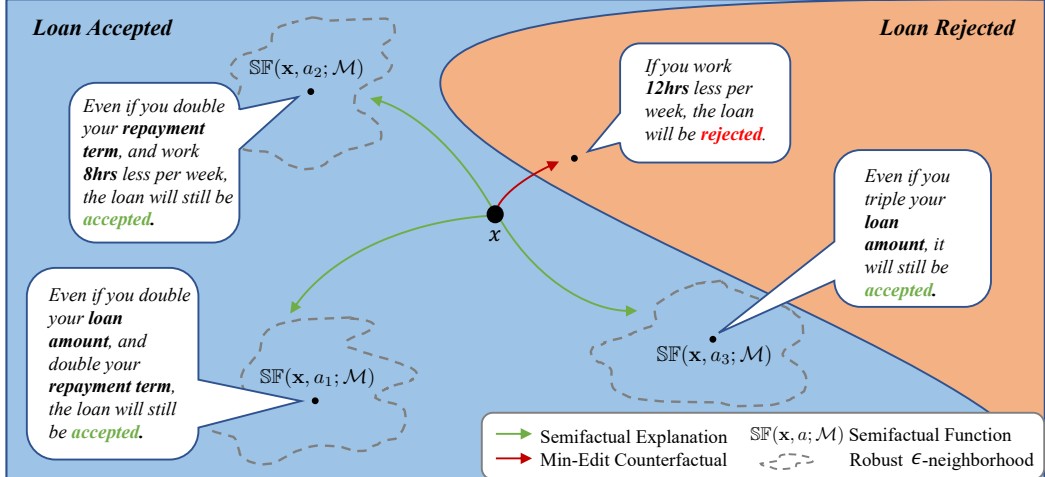

Figure 1: Semifactual Explanation to Optimise Positive Outcomes: An individual $\mathbf{x}$ has their loan accepted, but there are several semifactual explanations which can help optimise their outcome. Our algorithm produces a set of semifactual explanations which *maximise* the distance between $\mathbf{x}$ and the final explanation $\mathbb{SF}(\mathbf{x}, a; \mathcal{M})$. This allows the largest *Gain* to be achieved so that the user gets the maximum benefit. In contrast, counterfactual algorithms are not suitable because they are designed to target the shortest path across a decision boundary. In addition, the semifactuals are robust to distributional shifts by constraining an $\epsilon$-neighborhood between them and the decision boundary. Note $\mathcal{M}$ is the Structural Causal Model (SCM), see Section 3.

intervention of replacing $\mathbf{x}$ by $\boldsymbol{\theta}$ where $\boldsymbol{\theta} \in \mathcal{X}$. It implies that, for each feature, $X_i \coloneqq \theta_i$ for the individual $\mathbf{x}$. If the action $do(\boldsymbol{\theta})$ imposes no change, $\mathbf{x} = \boldsymbol{\theta}$ holds. We further denote a set of human-constrained actionable ranges $\mathcal{A} = \{a(\mathbf{x}) = do(\boldsymbol{\theta}) : \forall \boldsymbol{\theta} \in \mathcal{X}\}$. Note that the actions have to be mutable and explicitly exclude any action which keeps the individual in the same position.

The non-causal semifactual interaction between $\mathbf{x}$ and $a(\mathbf{x})$ is defined by $\mathbb{SF} : \mathcal{X} \times \mathcal{A} \mapsto \mathcal{X}$. That is, the individual $\mathbf{x}$ taking action $a(\mathbf{x})$ will lead to another representation $\boldsymbol{\theta} \in \mathcal{X}$ representing that person's recourse. Now, a structural semifactual is defined which considers the dependence between the related features [18, 24]. We denote the structural causal model (SCM) by $\mathcal{M} = (\mathbf{S}, \mathbb{P}_U)$ where $\mathbf{S}$ are a set of structural equations and $\mathbb{P}_U$ is the distribution over the exogenous variables $U \in \mathcal{U}$. Consider that in a causal graph, there is a set of causal parents for each feature $x_i \in \mathbf{x}$, denoted by $\mathrm{Pa}_i$. We denote the structural equations as $\mathbf{S} = \{x_i \coloneqq g_i(\mathrm{Pa}_i, U_i) : i = 1, \ldots, k\}$ where $g_i(\cdot)$ is a deterministic function that describes the causal relationship for $x_i$, and depends on the exogenous variable $U_i \in \mathcal{U}$ alongside the corresponding parent set $\mathrm{Pa}_i$. Hence, $\mathbf{S}$ induces a mapping $\mathbb{S} : \mathcal{U} \mapsto \mathcal{X}^*$ and its inverse mapping $\mathbb{S}^{-1} : \mathcal{X} \mapsto \mathcal{U}$. Let $f \circ g(x) = f(g(x))$ which can be extended to more functions. Hence, we specify the SCM-processed semifactual by $\mathbb{SF}(\mathbf{x}, do(\boldsymbol{\theta}); \mathcal{M})$ to denote the transition between the states by taking a certain action through an SCM $\mathcal{M}$, where

$$\boldsymbol{\theta}' = \mathbb{SF}(\mathbf{x}, do(\boldsymbol{\theta}); \mathcal{M}) \coloneqq \mathbb{S} \circ \mathbb{S}^{-1} \circ \mathbb{SF}(\mathbf{x}, do(\boldsymbol{\theta}); \mathcal{M}) . \tag{1}$$

If all features are *independently manipulable*, we have $\boldsymbol{\theta}' = \boldsymbol{\theta} = \mathbb{SF}(\mathbf{x}, do(\boldsymbol{\theta}); \mathcal{M}) = \mathbb{SF}(\mathbf{x}, do(\boldsymbol{\theta}))$. Therefore, $\mathbb{SF}(\mathbf{x}, do(\boldsymbol{\theta}); \mathcal{M})$ is a more generalised formulation which covers the non-causal case. Lastly, we assume a binary model that generates the score for the users is $h$, where $h : \mathcal{X} \mapsto \{0, 1\}$ by which we can simply consider that $1$ means a positive outcome (e.g., a loan acceptance) and $0$ is a negative outcome (e.g., a loan rejection). We set a lower threshold $\psi$ that separates the decision boundary. For the form defined above, $\psi = 0.5$ is a reasonable threshold that fits all situations well.

## 3.2 Framework

We define our semifactual framework as one centering on gain $(G)$ that is weighted by plausibility $(P)$, regularization in the form of diversity $(R)$, and hard constraints in the form of robustness $(H)$, indexed by $j$. All of the components are parameterized with $\mathbf{x}$ and a subset of suggestions $\{a_1, \ldots, a_m\}$. Letting $f(\cdot)$ be a function composed by gain and some weighting (i.e., plausibility for us), the causal semifactual framework is defined as

$$\max_{a_1, \ldots, a_m} \quad \frac{1}{m} \sum_{i=1}^{m} f(G(\mathbf{x}, a_i), P(\mathbf{x}, a_i)) + \gamma R(\{\boldsymbol{\theta}_1, \ldots, \boldsymbol{\theta}_m\})$$
$$\text{s.t.} \quad \boldsymbol{\theta}_i = \mathbb{SF}(\mathbf{x}, a_i; \mathcal{M}), H_j(\boldsymbol{\theta}_i) \geq 0, \forall i, j \tag{2}$$

where the regularisation and hard constraints can be multiple and indexed with $i$ and $j$, respectively. One may define a similar formulation for the non-causal case (see Section 4.2). We defer all details of the components until Section 3.4.

## 3.3 Optimising Positive Outcomes with *Gain*

For the core of the objective we appeal to the notion of gain. Note that *gain* is similar to the idea of *cost* commonly used in recourse [50], but there are three crucial differences. First, we are trying to *maximise* gain, rather than *minimise* cost [24]. Second, gain ideally considers the causal dependencies between features in its function, whilst cost typically only considers the user's action(s) [24]. Third, gain is further subdivided into positive and negative polarities. To elaborate on this last point, take the example of a user who has their loan application for buying a new house accepted. In this situation, if they desired to spend more time away from work with family, they would experience *positive gain* if they could work less hours per week and still have their loan accepted (see Figure 1). Conversely, if this person increased the number of hours they worked, they would experience *negative gain*. Notably, positive/negative gain is not necessarily connected to the model's probabilities (see $a_1$ in Figure 1 moving away from the decision boundary). Similar to actionability constraints which can offer individualised recourse [50], what is positive/negative gain must be manually defined for each individual. As prior work on semifactuals simply maximised the $L_2$ distance between a test instance and explanatory one to define good explanations [1, 29], we introduced the concept of gain to make them more meaningful in application.

More formally, we define the gain function by $G : \mathcal{X} \times \mathcal{A} \to \mathbb{R}$. By denoting $\boldsymbol{\theta} = \mathbb{SF}(\mathbf{x}, a; \mathcal{M})$, we decompose the function as follows:

$$G(\mathbf{x}, a) := \mathcal{P}_{SF} \circ \delta(\mathbf{x}, \boldsymbol{\theta}) = \mathcal{P}_{SF} \circ \delta(\mathbf{x}, \mathbb{SF}(\mathbf{x}, a; \mathcal{M})) \tag{3}$$

where $\mathcal{P}(\cdot, \cdot)$ is an oracle function that computes the payoff based on the vectorised difference between $\mathbf{x}$ and $\boldsymbol{\theta}$, i.e., $\delta : \mathcal{X} \times \mathcal{X} \mapsto \mathbb{R}^k$ which is a symmetrical difference function between the two feature representations. The subscript of $\mathcal{P}_{SF}$ denotes a semifactual. In interpretation, the gain function compares two states, (1) the original feature vector $\mathbf{x}$, and (2) the SCM-processed end state $\boldsymbol{\theta}$ which was led to through $\mathbf{x}$ taking action $a$.

**Why is *Gain* not necessarily equivalent to *Cost*?**  Formally, to enable the comparison, we write the cost function (denoted by $C(\mathbf{x}, a)$) as

$$C(\mathbf{x}, a) = -\mathcal{P}_{CF} \circ \delta(\mathbf{x}, \mathbb{SF}(\mathbf{x}, a)) \tag{4}$$

which builds on the fact that cost solely considers the feature change. Note that $\mathbb{SF}$ is equivalent to the notion $\mathbb{CF}$ in [25], what makes our approach different is the consideration of positive outcomes and gain. Our finding is that gain in semifactuals (SFs) is not necessarily equivalent to cost in counterfactuals (CFs) where the equivalence ignores the sign of both quantities, as formally stated as follows.

**Theorem 3.1.** *Even if $\mathcal{P}_{SF}(\cdot, \cdot) \equiv \mathcal{P}_{CF}(\cdot, \cdot)$, gain and cost are not necessarily equivalent ignoring the sign.*

*Proof.* Note that SCMs are also considered in counterfactual recourse [18, 24, 26]. However, in this prior research SCMs are typically applied for enforcing hard plausibility constraints, not in the computation of a user's cost. In contrast, our gain function takes the SCM-processed semifactual $\boldsymbol{\theta}'$ as an input. We employ proof by contradiction here. Assume that cost and gain are equivalent ignoring sign so that, without loss of generality,

$$|G(\mathbf{x}, a)| = |C(\mathbf{x}, a)| \iff |\mathcal{P} \circ \delta(\mathbf{x}, \mathbb{SF}(\mathbf{x}, a; \mathcal{M}))| = |\mathcal{P} \circ \delta(\mathbf{x}, \mathbb{SF}(\mathbf{x}, a))|$$
$$\iff \delta(\mathbf{x}, \mathbb{SF}(\mathbf{x}, a; \mathcal{M})) = \delta(\mathbf{x}, \mathbb{SF}(\mathbf{x}, a)) \tag{5}$$

holds. However, SCMs can result in possibly more features being changed since some features could be others' causal parents and those causal children will change their values accordingly. By denoting $\boldsymbol{\theta} = \mathbb{SF}(\mathbf{x}, a)$ and $\boldsymbol{\theta}' = \mathbb{SF}(\mathbf{x}, a')$, we consider the general case as follows:

$$|\delta(\mathbf{x}, \boldsymbol{\theta})| - |\delta(\mathbf{x}, \boldsymbol{\theta}')| = \sum_i |\delta(\mathbf{x}, \boldsymbol{\theta})_i| - \sum_i |\delta(\mathbf{x}, \boldsymbol{\theta}')_i|$$
$$= \sum_{\{i:\theta_i = \theta_i'\} \cup \{i:\theta_i \neq \theta_i'\}} |\delta(\mathbf{x}, \boldsymbol{\theta})_i| - |\delta(\mathbf{x}, \boldsymbol{\theta}')_i| = 0 + \sum_{\{i:\theta_i \neq \theta_i'\}} |\delta(\mathbf{x}, \boldsymbol{\theta})_i| - |\delta(\mathbf{x}, \boldsymbol{\theta}')_i| \leq 0, \tag{6}$$

which contradicts with Equation (5). Thus, even if the oracle function for calculating the payoff is the same, gain and cost are still not necessarily equivalent. Also, the equality in Equation (6) holds when all features are independently manipulable or the changed features are independently manipulable of the remaining features, so that $\mathbb{SF}(\mathbf{x}, a) = \mathbb{SF}(\mathbf{x}, a; \mathcal{M})$. The proof completes here. $\square$

### 3.4 Semifactual Components

Here, we detail how to incorporate the concepts of plausibility, robustness, and diversity into our framework for maximising gain, because they are agreed upon as important in the literature and useful for evaluation. While plausibility and diversity have been explored in semifactual explanation [1, 29], robustness and causality (and indeed an objective balancing all together) have not, yet we argue and show that the subtleties of "even if..." thinking are perhaps better captured in a casual setting.

**Plausible Gain**  We define plausibility here as explanations which are within distribution. For example, an explanation saying a person could earn less and still have their loan accepted should change their "debt-to-income ratio" feature also, or it will lie outside the data manifold. Prior work on semifactuals has only considered euclidean distance to training data as a heuristic for this [29], in contrast we posit (similar to the counterfactual literature [23]) that this is better approached with

SCMs. Hence, we define the plausibility for $\mathbf{x}$ taking the action $a$ by $P(\mathbf{x}, a) = \Pr(a = do(\boldsymbol{\theta})|\mathbf{x})$ where $\mathbf{x}$ is fixed for an individual and $\Pr(\cdot)$ is a density function. In our non-causal tests, we use the $L_2$ norm to training data to approximate plausibility (i.e., being in distribution, similar to [29, 32, 51]). However, this issue of plausibility is naturally taken care of in our causal tests thanks to the SCM ensuring plausible feature mutations, so we don't explicitly consider plausibility there going forward.

**Robust Gain**   Continuing with the example of a person who has a loan accepted to buy a house, the semifactual should sometimes be robust to distribution shifts. For example, if the person uses the semifactual explanation to triple their loan amount (recall Figure 1), they will likely need upwards of six months to locate a new house during which the semifactual should hold if the person e.g. gets an additional credit card. Hence, we define our semifactual robustness such that while taking action $a$, any close neighbor of the generated semifactual $\mathbb{SF}(\mathbf{x}, a; \mathcal{M})$ can still receive a positive outcome. The $\epsilon$-neighborhood of $\mathbf{x}$ centering around an individual $\mathbf{x}$ is

$$\mathcal{B}(\mathbf{x}) = \{\boldsymbol{\theta} = \mathbb{SF}(\mathbf{x}, a; \mathcal{M}) : \forall a \in \mathcal{A}, \delta(\boldsymbol{\theta}, \mathbf{x}) \leq \epsilon\} \tag{7}$$

which covers all neighbors that can be reached from $\mathbf{x}$ by taking an *actionable* feature change $a$ through the SCM $\mathcal{M}$. By definition, $\mathbf{x}$ is also a neighbor of itself since $\mathbf{x} \in \mathcal{B}(\mathbf{x})$ holds given $\delta(\mathbf{x}, \mathbf{x}) = 0$. Let us represent $\mathcal{B}(\mathbb{SF}(\mathbf{x}, a; \mathcal{M}))$ by $\mathcal{B}_s(\mathbf{x}, a)$ for simplicity. Given the predictive model $h(\cdot)$ and an individual $\mathbf{x}$, an action $a$ is robust for individual $\mathbf{x}$ if $h(\boldsymbol{\theta}) > \psi, \forall \boldsymbol{\theta} \in \mathcal{B}_s(\mathbf{x}, a)$, which is equivalent to $\min_{\boldsymbol{\theta} \in \mathcal{B}_s(\mathbf{x}, a)} h(\boldsymbol{\theta}) - \psi > 0$. For instance, $\psi = 0.5$ works for a binary model case. We hence denote the term related to the robustness by

$$H(\mathbf{x}, a) = \min_{\boldsymbol{\theta} \in \mathcal{B}_s(\mathbf{x}, a)} h(\boldsymbol{\theta}) - \psi \,, \tag{8}$$

which will be useful for constructing the final objective.

**Diverse Gain**   It is generally preferred to offer a number of suggested actions $\{a_1, \ldots, a_m\}$, rather than a single one [60]. Like prior work in counterfactuals, we define diversity as the average pair-wise distance among a set of entities [39, 61]. We reuse the distance function $\delta$ and define the diversity objective within a set of SFs $\{\boldsymbol{\theta}_i\}_{i=1}^m \subseteq \mathcal{X}^m$ as

$$R(\{\boldsymbol{\theta}_i\}_{i=1}^m) = \begin{cases} \frac{2}{m(m-1)} \sum_{i=1}^m \sum_{j>i}^m L_2 \circ \delta(\boldsymbol{\theta}_i, \boldsymbol{\theta}_j) & m > 1 \\ 0 & m = 1 \end{cases} \tag{9}$$

which represents a pairwise mean distance among the set of data points, based on the $L_2$ norm. One may accommodate $m = 1$ for the case when only a single semifactual is desired.

### 3.4.1   Semifactual Objective

The final objective may be constructed as follows.

**Definition 3.2** (Semifactual Objective). We consider a simple composition multiplication function for $f(\cdot)$. Considering gain, plausibility, robustness, and diversity, the semifactual objective function is:

$$\max \quad \frac{1}{m} \sum_{i=1}^m P(\mathbf{x}, a_i) G(\mathbf{x}, a_i) + \gamma R\left(\{\mathbb{SF}(\mathbf{x}, a_i; \mathcal{M})\}_{i=1}^m\right) \tag{10}$$

$$\text{s.t.} \quad \forall i = 1, \ldots, m : a_i \in \mathcal{A}, H(\mathbf{x}, a_i) > \psi \,.$$

In optimisation [10], an adversarial interpretation from the perspective of a two-player zero sum game can further simplify Equation (10) to

$$\mathcal{J} := \min_{\lambda_1, \ldots, \lambda_m \geq 0} \max_{a_1, \ldots, a_m \in \mathcal{A}} \frac{1}{m} \sum_{i=1}^m P(\mathbf{x}, a_i) G(\mathbf{x}, a_i) + \lambda_i H(\mathbf{x}, a_i) + \gamma R\left(\{\mathbb{SF}(\mathbf{x}, a_i; \mathcal{M})\}_{i=1}^m\right) \,, \tag{11}$$

where $H(\mathbf{x}, a)$ is the Lagrangian. The primal player tries to maximise the plausibility-weighted gain and diversity, with regard to $a$, whilst the dual player tries to minimise regarding a set of $\lambda$.

Since there are $m$ suggestions, the constraints for robustness will be $m$ times. Observing the objective, the robustness is a hard constraint, whilst the diversity can be regarded as regularisation. $P$ can be seen as a scaling factor for $G$ which helps to guarantee that high expected gain is only possible alongside high plausibility, simply adding them misses this special property.

## 3.5 Properties of the Framework

**Effective Solution Space.** We discuss the set of meaningful solutions here and the result validates the re-formulation [i.e., Equation (11)] of the semifactual framework [i.e., Equation (10)]. First, we depict the lemma.

**Lemma 3.3.** *Assume that the limit of the gain function and diversity term are finite. Also, assume that $\mathcal{A}^+ := \{a \in \mathcal{A} : G(\mathbf{x}, a) \geq 0\}$ is non-empty for an individual $\mathbf{x}$. The semifactual objective $\mathcal{J} \geq 0$ when $\forall i = 1, \ldots, m, a_i \in \mathcal{A}^+ : H(\mathbf{x}, a_i) \geq 0$, otherwise $\mathcal{J} = -\infty$.*

See Section A.1 for the proof. We can summarise that the action set which is able to provide the positive payoff can be defined by the named effective solution space for $\mathbf{x}$: $\mathcal{A} = \{a \in \mathcal{A} : H(\mathbf{x}, a) \geq 0, G(\mathbf{x}, a) > 0\}$. Hence, repeated suggestions will be produced when the number of actions in this space is smaller than the required $m$. Otherwise, the solution will provide more versatile options. There are no suggestions to achieve an effective semifactual(s) (i.e., with positive gain) if this solution space is empty. However, similar situations exist for counterfactuals when they are also impossible to generate, assuming a similar set of actionable constraints are defined.

# 4 Implementation Details

We now introduce our methods to solve Equation (11), henceforth called Semifactual-recourse GENeration (S-GEN), for both causal and non-causal domains. In the following paragraphs, we use $\hat{G}$ to denote an empirical approximation of $G$, and likewise for $P$, $H$, $R$, and $\mathcal{J}$.

## 4.1 Causal Case

Assuming the presence of a differentiable classifier $h(\cdot)$ and SCM $\mathcal{M}$, (recall the latter guarantees plausibility), let $\Omega_i(\mathbf{x}) = \{\Pr(\boldsymbol{\theta}) : \boldsymbol{\theta} \in \mathcal{B}_s(\mathbf{x}, a_i)\}$ be the probability distribution over the $\epsilon$-neighborhood of $\mathbb{SF}(\mathbf{x}, a_i; \mathcal{M})$. Also, let $\mathbf{B}_i$ represent a finite subset of $\mathcal{B}_s(\mathbf{x}, a_i)$ sampled according to $\Omega_i(\mathbf{x})$. Our objective is:

$$\max_{a_1,\ldots,a_m} \min_{\lambda_1,\ldots,\lambda_m} \quad \frac{1}{m} \sum_{i=1}^m -\lambda_i \mathcal{L}\left(h(\mathbb{SF}(\mathbf{x}, a_i; \mathcal{M}), h(\mathbf{x})\right) - \frac{1}{|\mathbf{B}_i|} \sum_{\boldsymbol{\theta}_i \in \mathbf{B}_i} \lambda_i \mathcal{L}\left(h(\boldsymbol{\theta}_i), h(\mathbf{x})\right)$$
$$+ \hat{P}(\mathbf{x}, a_i)\hat{G}(\mathbf{x}, a_i) + \gamma \hat{R}(\{\mathbb{SF}(\mathbf{x}, a_i; \mathcal{M})\}_{i=1}^m)$$
$$\text{s.t.} \quad \forall i = 1, \ldots, m : a_i \in \mathcal{A}, \lambda_i > 0 \tag{12}$$

where $\mathcal{L}$ is the binary cross entropy loss. For robustness, we used Monte Carlo (MC) sampling with an epsilon $\epsilon$ robust hypersphere, and if either the instance or sampling return a *negative outcome* with $h(\cdot)$, we use the prior optimisation step as the solution. For diversity, $m$ is set to the number of actionable feature sets, and a solution is obtained for each. We utilise the causal recourse approach of Karimi et al. [26] for solving the maximin. The actionable bounds are clipped each iteration, and $\lambda$ is iteratively decreased to put more emphasis on gain over time (see Algorithm 2).

## 4.2 Non-Causal Case

For the non-causal case, we use a genetic algorithm [53, 58] which only assumes a binary predictive model $h(\cdot)$. This approach follows the standard design for genetic algorithms, with some minor alterations specifically for semifactual generation, see Appendix D for the pseudocode. Next we present the fitness function which optimises our objective.

### 4.2.1 Fitness Function

For gain, the average distance between an individual $\mathbf{x}$ and each semifactual $\mathbb{SF}(\mathbf{x}, a)$ is measured as $\hat{G}(\mathbf{x}, a) = \|\mathbb{SF}(\mathbf{x}, a) - \mathbf{x}\|_2$. For robustness, we relax it to two constraints: $H_p$ is the probabilistic robustness for the neighbor points where the generated semifactuals for a query are randomly perturbed using MC simulation to make sure the surrounding neighborhood is robust, and $H_a$ the absolute robustness for the individual $\mathbf{x}$ (more detail in Section A.2). For the first constraint, a score of $\hat{H}_p(\mathbf{x}, a) = \frac{1}{n} \sum_{i=1}^n \mathbb{1}\{h(\mathbf{x}) = h(\boldsymbol{\theta}_i)\}$ where $\boldsymbol{\theta} \sim \Pr(\mathcal{B}_s(\mathbf{x}, a))$, is returned. For

the second constraint, a score of $\hat{H}_a(\mathbf{x}, a) = \mathbb{1}\{h(\mathbf{x}) = h \circ \mathbb{SF}(\mathbf{x}, a)\}$ is returned. Hence, the solution is rewarded for (1) the neighborhood samples, and (2) the semifactuals themselves being classified as $h(\mathbf{x})$. For plausibility, we take from prior work and directly use the training data [29]. Specifically, considering the training data set $\mathcal{D}$, we define the notion of plausibility using the distance of each semifactual generated to the nearest training data point. As the term must be maximised, we use a function which is monotonically decreasing with respect to the distance with $P(\mathbf{x}, a) \approx \hat{P}_\mathcal{D}(\mathbf{x}, a) = \exp\{1/(\min_{\boldsymbol{\theta} \in \mathcal{D}} \|\mathbb{SF}(\mathbf{x}, a) - \boldsymbol{\theta}\|_2^2 + \gamma_p)\}$ where $\gamma_p$ is to account for when a perfect match to the semifactual exists in the training data (thus the division is undefined), and $\hat{P}_\mathcal{D}(\mathbf{x}, a)$ is an empirical approximation (based on $\mathcal{D}$) of the plausibility. Lastly, for diversity [39, 61], we take the mean distance between all $m$ generated semifactuals with $\hat{R}(\{\mathbb{SF}(\mathbf{x}, a_i)\}_{i=1}^m)$ which precisely follows Equation (9). This objective collapses to 0 when $m = 1$.

Certain objectives need to be weighted individually based on the problem. For example, explanations which can be acted upon immediately perhaps don't need robustness. Notably, the multiplier $\lambda$ in Equation (11) is split to $\lambda_p$ and $\lambda_a$, for $H_p$ and $H_a$ respectively. In this work, we treat them as hyperparameters. Also, they are used alongside $\gamma$ to balance the objectives. Since $\lambda$ and $\gamma$ are selected as hyperparameters, they are removed under the min operator. Finally, the objective (fitness) function is defined as:

$$\max_{a_1,...,a_m \in \mathcal{A}^+} \frac{1}{m} \sum_{i=1}^m \hat{P}_\mathcal{D}(\mathbf{x}, a_i) \hat{G}(\mathbf{x}, a_i) + \lambda_p \hat{H}_p(\mathbf{x}, a_i) + \lambda_s \hat{H}_a(\mathbf{x}, a_i) + \gamma \hat{R}(\{\mathbb{SF}(\mathbf{x}, a_i; \mathcal{M})\}_{i=1}^m).$$

We selected the hyperparameters via a grid search, see Appendix C. Crucially, we also weight the fitness function output by $\hat{H}_p(\mathbf{x}, a)$ to encourage solutions with more semifactuals (see Algorithm 1).

## 5   Experiments & Results

Here we test S-GEN in both causal and non-causal settings. We show the effectiveness of our method in optimising a user's positive outcome compared to baselines and open source our code (see Appendix E). The actionability constraints are detailed in Appendix B. Baselines were modified to be appropriately compared, most importantly, we stopped counterfactual techniques before they crossed a decision boundary (thus generating semifactuals), and modified semifactual techniques to work on tabular data, Appendix G details the peripheral modifications.

In the non-causal setting, we consider three datasets, Loan Application [33], German Credit [21], and BCSC [11]. All categorical variables are one hot encoded. Three models were used, a decision tree, logistic regression, and naïve bayes, each with 30 random test data point explanation samples gotten by varying the random seed. Note that because the range varied on each dataset, the results were normalised and averaged for each, but Appendix F details each individual dataset for completeness. For baselines, we modify three techniques, DiCE by Mothilal et al. [39] (henceforth DiCE*), PIECE by Kenny & Keane [29] (henceforth PIECE*), and Diverse semifactual Explanations of Reject by Artelt & Hammer [1] (henceforth DSER*). Plausibility is measured as the distance between a generated semifactual(s) and the nearest training example; thus, the smaller the better. Robustness is measured by MC sampling $n = 100$ single feature perturbations of each semifactual $\boldsymbol{\theta}_i$, predicting their class, and returning a float between 0-1 of the success rate as described in Section 4.2.1.

In the causal setting, the Adult [31] and COMPAS [5] datasets are considered. The SCMs from Nabi & Shpitser [41] were used, and the structural equations from Dominguez et al. [18]. All categorical features are treated as real-valued. We use the pre-trained MLP classifiers from Dominguez et al. [18] and take 30 averaged samples from 5 random seeds. As baselines we modify the technique of Karimi et al. [26] [henceforth Karimi et al.(2021)*], and Dominguez et al. [18] [henceforth Dominguez et al.(2022)*], the latter optimises with robustness in mind. We optimise the relevant techniques to be robust in an $\epsilon = 0.1$ hypersphere, and the C&W adversarial attack by Carlini & Wagner [12] measures robustness by checking if the nearest adversarial attack is outside this radius.

For all tests, the main metric of concern is gain, that is, the mean distance between a query and its generated semifactual(s), the larger this number, the better. Diversity is also measured for all tests as the mean distance between all $m$ generated semifactuals for an individual $\mathbf{x}$, the higher the number, the better. To be in line with prior art, the $L_2$ norm is used in non-causal tests [1], and the $L_1$ in causal [18]. Note for causal tests the SCM guarantees plausibility so this metric is not reported.

## 5.1 Non-Causal Results

Our purpose here is to show that current methods are insufficient to meet the basic requirements for semifactual explanation discussed in Section 3.4. Specifically, a technique needs to optimise gain, while remaining plausible, robust, and offering diverse explanations.

Observing the average normalised results across all datasets (note robustness was not normalised since it is already 0-1 range), Figure 2 shows that S-GEN performed the best on all metrics for all values of $m$ (1-10). The results demonstrate that traditional counterfactual approaches (DiCE*) are not suitable to achieve optimal gain, due to them focusing on minimising cost. Moreover, methods built for semifactual generation specifically (i.e., DSER* & PIECE*) that *do* actually maximise gain somewhat, still fail to match the results of S-GEN. This shows that S-GEN is superior to existing semifactual methods (and popular counterfactual approaches appropriately modified) for maximising a user's gain in positive outcomes. Moreover, it does so while maintaining superior plausibility, robustness, and diversity in all tests.

## 5.2 Causal Results

We evaluate our algorithm in a causal setting where the SCMs and structural equations are known. The primary purpose of this test is to demonstrate that the semantics of semifactual "even if" thinking is better captured in a causal setting due to dependencies being taken into account when calculating a person's gain. With regard to diversity, we fix $m$ to the maximum number of feature sets available from the actionable features (so only one $m$ value is tested).

Figures 3a and 3b show the initial gain achieved by a person after taking a certain action (i.e., the *Action Gain*), and how this gain transforms after considering the causal relationship between features (i.e., the *Causal Gain*). Firstly, the total gain achieved by S-GEN is much larger than the baselines in both datasets and hence consistent with our non-causal tests. More importantly however, the change in gain a person achieves after considering the causal relations in the adult dataset is significantly higher both in significance testing and effect size ($0.055 \pm 0.001$ v. $0.063 \pm 0.001$; t-test $p < 0.02$; Cohen's $d = 2.24$), showing it is beneficial to consider causality when calculating a person's gain. The results of diversity put S-GEN first also (S-GEN $= 0.84 \pm 0.09$ v. Karimi $= 0.43 \pm 0.03$ v. Domineguez $= 0.34 \pm 0.03$). In robustness, both S-GEN and Domineguz et al. (2022)* did reasonably well (S-GEN $= 87\%$ success v. Domineguz $= 54\%$ success), but Karimi et al. (2021)* did not (7.2% success), likely due to the latter not being designed for this.

## 6 User Evaluation

The primary motivation behind this work is the hypothesis that semifactual explanation would be preferred by users over counterfactuals in positive outcome settings. To test this assumption, we design the first user test in XAI directly comparing the two. Specifically, we show users three materials in which a person has a bank loan accepted, and three in which they don't. Users were then shown both explanation types for each material, and asked to rate on a scale from 1-5 how useful each were. So, the study was a within-subjects design, and the condition was the explanation type. Note that although we are studying the effect of the explanation type on loan acceptance, the

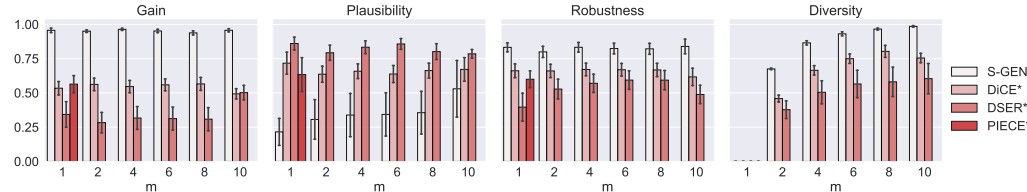

Figure 2: Results: The ability of S-GEN to create semifactuals is compared to DiCE*, DSER*, and PIECE*. Overall, S-GEN does the best, achieving significantly better results to all baselines in all tests. Note we normalised all results before averaging because each dataset has different scaling. Standard error bars are shown.

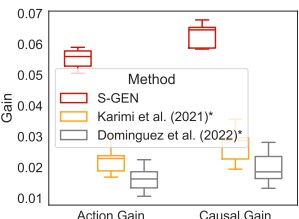

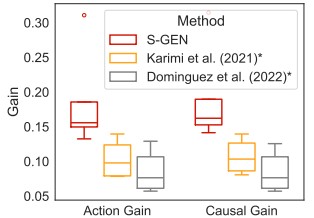

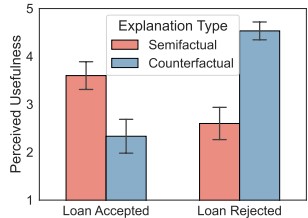

| (a) Causal Experiment: Adult | (b) Causal Experiment: COMPAS | (c) User Study |
|---|---|---|

Figure 3: Causal Experiment & User Study Results: (a/b) show the gain achieved by all methods both before and after considering the causal dependencies. Firstly, note that S-GEN achieves significantly more gain than the alternatively proposed approaches. Most importantly however, (a) shows there is significantly more gain achieved on the Adult data by S-GEN after taking causal dependencies into account, showing the importance of a causal formalisation. (c) Shows the user study results, where people perceive semifactual explanation as being significantly more useful than counterfactuals in the positive outcome of having a loan accepted. Standard error bars are shown.

loan rejection scenarios were also included to balance people's view of the problem setting, and as attention checks to verify that users were engaging with the materials and varying their scores accordingly. For analysis, each user's scores for counterfactuals and semifactuals were averaged in both loan acceptance and rejection materials into four decimal scores per user, thus allowing us to analyse the discrete Likert scores with t-tests [28]. As is a popular approach [22], we don't explicitly define what "useful" means to users, but rather let them use their own natural interpretation, as the results returned were reasonably consistent across individuals, they appear to have converged on an common interpretation of this word. The null hypothesis is that people will find both explanation types not significantly different in loan acceptance. The alternative is that people will find semifactuals significantly more useful in loan acceptance.

A power analysis [13] of two dependent means with an effect size $dz = 0.8$, alpha $\alpha = 0.05$, and power $(1 - \beta$ err prob)=0.9 informed a sample of 15 was appropriate for t-tests. Users were gathered from Prolific.com, 8 males, 7 females, aged 18+, native English speakers, and from the U.S. People were paid \$12/hr, which totalled \$35. The semifactuals were generated with S-GEN, and the counterfactuals with DiCE [39], notably these are equivalent to *positive counterfactuals* by McGrath et al. [36] for explaining loan acceptance situations. The study obtained IRB approval from MIT.

All users engaged and changed their ratings significantly depending on whether a loan was accepted or rejected, so none were excluded. Figure 3c shows users find semifactuals significantly more useful in loan acceptance (S-GEN=3.60±0.27 v. DiCE=2.33±0.34; $p < .005$) compared to rejections when counterfactuals are preferred (S-GEN=2.6±0.32 v. DiCE=4.53±0.17; $p < .0001$). Hence we reject the null and lend credible evidence that semifactuals are more useful to explain positive outcomes.

## 7  Discussion

Although much XAI work has explored how to explain positive outcomes, to the best of our knowledge, no consideration has been given towards explaining how to *optimise* them. Here, we have taken the novel step of exploring this, and showed how semifactuals are especially suited for the purpose. This required building on prior work in semifactuals by (1) introducing the concept of *Gain*, (2) re-framing them in a causal setting, and (3) conducting their first user test in XAI. Perhaps the notable limitation of our work is that although we have shown people do perceive semifactuals as being more useful in positive outcomes, we have not demonstrated this quantitatively, notably because of the difficulties acquiring an appropriate user base alongside the ethical considerations of such a study. Moreover, considering a casual formulation of semifactuals requires an SCM, which is not always realistic, but we have provided a non-causal algorithm for these situations. In future work, it would be interesting to formalise the utility of semifactuals for optimising positive outcomes in other domains such as robotics, which likely requires other considerations.

## Acknowledgements

The authors would like to thank Neil J. Hurley, alongside Mark T. Keane and Ruth M.J. Byrne who both inspired early consideration of the ideas in this paper. The authors would also like to thank MIT for their support in this project. This research wasn't directly supported by any grants or funding. We hope readers find the ideas interesting and useful for application.

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

# A  Property Analysis

## A.1  Proof of Lemma 3.3

*Proof.* We can rewrite $\mathcal{J}$ to

$$\mathcal{J} = \max_{a_1,\ldots,a_m \in \mathcal{A}} \left\{ \frac{1}{m} \sum_{i=1}^{m} P(\mathbf{x}, a_i) G(\mathbf{x}, a_i) + \frac{1}{m} \sum_{i=1}^{m} \min_{\lambda_i \geq 0} \lambda_i H(\mathbf{x}, a_i) + \gamma R\left(\{\mathbb{SF}(\mathbf{x}, a_i; \mathcal{M})\}_{i=1}^{m}\right) \right\}. \tag{13}$$

We derive the fact that, for any $i$,

$$\min_{\lambda_i \geq 0} \lambda_i R(\mathbf{x}, a_i) = \begin{cases} -\infty & H(\mathbf{x}, a_i) < 0 \\ 0 & \text{otherwise} \end{cases}$$

where $-\infty$ comes from setting $\lambda_i = \infty$ and $0$ is obtained by setting $\lambda_{p_i} = 0$. By the linearity of summation, we can further derive

$$\frac{1}{m} \sum_{i=1}^{m} \min_{\lambda_i \geq 0} \lambda_i R(\mathbf{x}, a_i) = \begin{cases} -\infty & \exists i, H(\mathbf{x}, a_i) < 0 \\ 0 & \text{otherwise} \end{cases}.$$

That is, if any constraint for the robustness is unsatisfied, the dual player will minimise the objective towards $-\infty$; however, the primal player cannot optimise towards $\infty$ given that the limit of the gain function and the diversity are finite. In other words, if the constraints are satisfied, the primal player can freely optimise the objective. Once $H(\mathbf{x}, a_i) \geq 0, \forall a_i$ are satisfied, the objective becomes

$$\tilde{\mathcal{J}} := \max_{a_1,\ldots,a_m \in \mathcal{A}^+} \frac{1}{m} \sum_{i=1}^{m} P(\mathbf{x}, a_i) G(\mathbf{x}, a_i) + \gamma R\left(\{\mathbb{SF}(\mathbf{x}, a_i; \mathcal{M})\}_{i=1}^{m}\right)$$

$$\geq \min_{a_1,\ldots,a_m \in \mathcal{A}^+} \frac{1}{m} \sum_{i=1}^{m} P(\mathbf{x}, a_i) G(\mathbf{x}, a_i) + \gamma R\left(\{\mathbb{SF}(\mathbf{x}, a_i; \mathcal{M})\}_{i=1}^{m}\right) \tag{14}$$

$$> 0$$

as $P(\mathbf{x}, a_i) > 0$ and $G(\mathbf{x}, a_i) \geq 0$ for any $a_i \in \mathcal{A}^+$; also, $R \geq 0$ holds. We conclude the proof here. $\square$

## A.2  A Probabilistic Relaxation of Robustness

Absolute robustness is difficult to guarantee, and common practice is to relax this via a probabilistic approach [26].

Assume there is a distribution over the sample space $\mathcal{B}_s(\mathbf{x}, a)$ denoted by $\Pr(\mathcal{B}_s(\mathbf{x}, a))$. We write $\boldsymbol{\theta} \sim \Pr(\mathcal{B}_s(\mathbf{x}, a))$ to indicate that $\boldsymbol{\theta}$ is sampled from the set $\mathcal{B}_s(\mathbf{x}, a)$ under the density $\Pr(\cdot)$. Let $\mathbb{E}[h(\boldsymbol{\theta})|\mathbf{x}, a]$ denote the expectation of $\boldsymbol{\theta}$ in this configuration. Hence, we modify Equation (8) to

$$\mathbb{E}[h(\boldsymbol{\theta})|\mathbf{x}, a] > \tilde{\psi}, \tag{15}$$

where $\tilde{\psi}$ is a function that adjusts the base score threshold $\psi$. It is crucial to have this threshold function in order to consider the variance of scores in the neighbor set. Particularly, we would like most neighbors to remain in a similarly "good" state, with low variance between them.

Moreover, we explicitly impose $h(\mathbb{SF}(\mathbf{x}, a; \mathcal{M})) - \psi > 0$. It places a hard constraint to avoid the case in which the neighbors of the semifactual are robust, but the "semifactual" itself has crossed the decision boundary to become a counterfactual. Whilst somewhat unlikely, this situation is theoretically possible, and requires consideration. In this case, $H$ is re-written as $H_p$, which represents a combination of (i) the probabilistic robustness, and (ii) the absolute robustness for the semifactual $H_a$ such that:

$$H_p(\mathbf{x}, a) = \mathbb{E}[h(\boldsymbol{\theta})|\mathbf{x}, a] - \tilde{\psi} \qquad H_a(\mathbf{x}, a) = h(\mathbb{SF}(\mathbf{x}, a; \mathcal{M})) - \psi. \tag{16}$$

In practice, $H_p$ is still non-trivial to solve. Monte Carlo (MC) sampling is a common strategy to apply here such that, by sampling a fixed sized batch $\mathbf{B} = \{\boldsymbol{\theta} : \boldsymbol{\theta} \sim \Pr(\mathcal{B}_s(\mathbf{x}, a))\}$,

$$H_p(\mathbf{x}, a) = \mathbb{E}[h(\boldsymbol{\theta})|\mathbf{x}, a] - \tilde{\psi} \approx (1/|\mathbf{B}|) \sum_{\boldsymbol{\theta} \in \mathbf{B}} h(\boldsymbol{\theta}) - \tilde{\psi}. \tag{17}$$

This implies that we substitute an unbiased estimator for the population mean.

# B Actionability Constraints

## B.1 Non-Causal

Here we define the actionability constraints used in the various domains. It may be assumed that the direction features are allowed to change corresponds with *positive gain*. We use various sized "action sets" to fully test all algorithms in various setups. The German Credit data used 15 actionable features to be closely in line with Mothilal et al. [39] whom allowed all features to be mutable. However, we also used 7 on Lending Club, and 4 on Adult Census/Breast Cancer to test the algorithms in situations with smaller action spaces also for completeness.

We ordered categorical features in a sensible fashion to "direct" semifactual "even if" thinking, and when we say a categorical feature could decrease/increase, we are referring to this pre-defined order. If you are interested in the exact ordering, please refer to our code which contains all the lists, but here we summarise. In reality however, a user must specify their exact actionability constraints, what we have specified here is designed to be representative what is possible for the "average" individual.

### B.1.1 German Credit Dataset

The continuous features used were 'duration', 'amount', 'age', the categorical ones were 'status', 'credit_history', 'purpose', 'savings', 'employment_duration', 'installment_rate', 'personal_status_sex', 'other_debtors', 'present_residence', 'property', 'other_installment_plans', 'housing', 'number_credits', 'job', 'people_liable', 'telephone', 'foreign_worker'. As actionable features for semifactual recourse, we considered the following:

- *duration*: We allowed people to increase the duration of their loan.
- *amount*: We allowed people to increase the amount of their loan.
- *status*: We allowed people to move towards having lower status.
- *credit_history*: We allowed people to move towards e.g. having a late payment if their credit history was otherwise good.
- *savings*: This feature was allowed to decrease.
- *employment_duration*: This feature was allowed to decrease in case people wanted to e.g. start a new job.
- *installment_rate*: This feature was allowed to move towards lower payments.
- *other_debtors*: this feature was allowed to add another co-applicant.
- *present_residence*: This feature was allowed to move towards e.g. renting in case the user desired to do so whilst searching for a new house with their loan.
- *property*: this feature was allowed to move towards having no property in case the user desired to sell their house/car etc to help pay for e.g. a downpayment.
- *other_installment_plans*: This feature was allowed to add other installment plans.
- *housing*: this feature was allowed to move towards renting away from e.g. owning.
- *number_credits*: This feature was allowed to increase if the user desired to acquire more credit cards.
- *job*: this feature was allowed to decrease in case the individual desired to get a different, less demanding job within their institution, or indeed quite their job to e.g. start a business.
- *people_liable*: This feature was allowed to move towards more people being liable.

### B.1.2 Lending Club

The continuous features used were 'loan_amnt', 'pub_rec_bankruptcies', 'annual_inc', 'dti', the categorical ones were 'emp_length', 'term', 'grade', 'home_ownership', 'purpose'. As actionable features for semifactual recourse, we considered the following:

- *home_ownership*: This feature was allowed to decrease towards e.g. renting.
- *annual_inc*: this feature was allowed to decrease if the person desired to e.g. work less hours.

- *emp_length*: This feature was allowed to decrease in case the individual desired to change careers.
- *dti*: dept to income ratio, this feature was allowed to increase.
- *pub_rec_bankruptcies*: This feature was allowed to increase in case the user decided they wanted to declare bankruptcy to e.g. try and keep some assets.
- *loan_amnt*: this feature was allowed to increase.
- *term*: This feature was allowed to decrease.

### B.1.3   Breast Cancer

The continuous features used were none, the categorical ones were 'agegrp', 'density', 'race', 'Hispanic', 'bmi', 'agefirst', 'nrelbc', 'brstproc', 'lastmamm', 'surgmeno', 'hrt'. As actionable features for semifactual recourse, we considered the following:

- *bmi*: This feature was allowed to move towards less healthy BMI levels in case the patient e.g. has hypothyroidism.
- *brstproc*: this feature was allowed to move towards having had a previous breast proceedure in case the patient would like to do so or was advised.
- *hrt*: This feature was allowed to move towards starting HRT, in case a person may wish to alleviate synthoms of the menopause.
- *agegrp*: this feature was allowed to get older in case the individual would like to take no action confident that it would not lead to cancer in the next few years/decades.

## B.2   Causal

In the causal setting, we allowed a user's age to increase a maximum of 5 years to mimic the motivating examples in the paper about a user having a bank loan accepted. In such a situation, the user may want to e.g. work less hours over the next 5 years whilst they repay the loan, and still have it accepted.

Next, we detail the direction features are allowed to change, and what direction corresponds to *positive gain*.

### B.2.1   Adult Income Census

We use the features "sex", "age", "native-country", "marital-status", "education-num", "hours-per-week", which are the variables in the causal graph of Nabi & Shpitser [41]. We consider "age" and "hours-per-week" as actionable. We allow "age" to increase a maximum of five years, and "hours-per-week" to decrease.

For positive gain, we considered: Age, marital status, and eduation-num *increasing* corresponding to positive gain, and hours-per-week *decreasing* corresponding to positive gain. A persons sex was seen as neutral gain.

### B.2.2   COMPAS

We use the features "age", "race", "sex" and "priors count", which are the variables in the causal graph of Nabi & Shpitser [41]. We consider "age" and "priors count" as actionable. As actionability constraints, we assume that both features are non-negative and can only be increase. Age specifically is only allowed to increase by 5 years for each individual.

For positive gain, we considered: Age and priors count increasing corresponding to positive gain. A persons sex and race was seen as neutral gain.

## C   Hyperparameter Choices

In this section, we discuss the hyperparameter specifications for the causal and non-causal cases respectively.

## C.1 Non-Causal

Here we note the values for the hyperparameters used in our demonstrations. All were obtained though pilot grid-searches across each dataset. The hyperparameter choices are summarised in Table 1

Table 1: Hyperparameter Specifications

| Data | $\lambda_p$ | $\lambda_s$ | $\gamma_d$ | $\gamma_p$ |
|---|---|---|---|---|
| German credit | 30 | 10 | 1 | $1e^{-1}$ |
| Lending Club | 30 | 10 | 1 | $1e^{-1}$ |
| Breast Cancer | 10 | 10 | 10 | $1e^{-1}$ |

For S-GEN itself, we used the same hyperparameters everywhere outside of the above table. The number of generations spent searching for a solution was 20. The population size was fixed at $\{12, 24, 48, 72, 96, 120\}$, for diversity sizes of $\{1, 2, 4, 6, 8, 10\}$, respectively. The mutation rate was 0.05. The number of "elite" solutions passed on for each generation was 4. The probability of a crossover happening was 0.5. The number of Monte Carlo trials for each instance was 100. The continuous features were perturbed (in mutation or population initialization) by the output from sampling a standard normal distribution with standard deviation equal to the max actionable feature value, minus the min actionable feature value, multiplied by 0.05.

## C.2 Causal

In our causal tests we chose $\lambda$ as 1.0, and this was gradually decreased by a momentum of $\eta$=0.9 each iteration to put more emphases on the maximization of gain.

# D  Algorithm Pseudocode

---

**Algorithm 1** S-GEN: Genetic Algorithm to Generate semifactual Recourse with Robustness and Diversity in a Non-Causal Model Agnostic Setting

---

**Require:** $\mathbf{x}$ the user feature
**Require:** $h(\cdot)$ the predictive model
**Require:** $m$ the expected number of suggestions
**Require:** $n$ the number of candidates, $n > m$
**Ensure:** $\mathbf{R}_{SF}$ the set of semifactual(s)
 1: Sample $n$ candidates $\mathbf{D} \leftarrow \{\boldsymbol{\theta}_i \sim \mathcal{X}\}_{i=1}^n$
 2: **while** the stopping criterion is not satisfied **do**
 3:     Obtain the fitness scores $\mathbf{f}$ with respect to $\mathbf{D}$
 4:     Save the fittest $\boldsymbol{\theta}^* \in \mathbf{D}$ according to $\mathbf{f}$
 5:     Let $\mathbf{D}$ evolve by *natural selection* according to $\mathbf{f}$, *crossover*, *mutation*, and *elitism* with $\mathbf{x}^*$
 6: **end while**
 7: Collect the best $m$ unique candidates from $\{\boldsymbol{\theta} \in \mathbf{D} : h(\boldsymbol{\theta}) = h(\mathbf{x}) = 1\}$ to $\mathbf{R}_{SF}$, according to the corresponding fitness scores in $\mathbf{f}$
 8: **if** $|\mathbf{R}_{SF}| < m$ **then**
 9:     Complement $\mathbf{R}_{SF}$ to $m$ elements with $\boldsymbol{\theta}$ randomly drawn from $\mathbf{R}_{SF}$
10: **end if**

---

# E  Code

For our full code used please see:

https://github.com/EoinKenny/Semifactual_Recourse_Generation

The ability to reproduce the results is given.

**Algorithm 2** S-GEN: Algorithm to Generate Robust & Diverse Causal semifactual Explanations for Differentiable Classifiers

---

**Require:** $\mathbf{x}$ the user feature vector
**Require:** $h(\cdot)$ the predictive model
**Require:** $\mathcal{M}$ the **differentiable** SCM
**Require:** $\epsilon$ the epsilon robustness
**Require:** $\eta$ the momentum parameter
**Require:** $\tau$ the learning rate
**Require:** $\mathrm{Proj}(\cdot)$ a projection function that ensures the action is actionable
**Ensure:** $\mathbf{R}_{SF}$ the set of semifactual(s)
 1: $\mathbf{R}_{SF} \leftarrow \emptyset$
 2: $i \leftarrow 0$
 3: **for** $a \in \mathcal{A}$ **do**
 4:    Move to next loop if the SF generated with the initial $a$ does not satisfy the constraints.
 5:    $a_i \leftarrow a$
 6:    **while** not converged **do**
 7:      Sample a batch of neighbors from $\mathbb{B}_s(\mathbf{x}, a_i)$, denoted by $\mathbf{B}_i$
 8:      **if** $h\big(\mathbb{SF}(\mathbf{x}, a_i; \mathcal{M})\big) = 0$ or $h(\boldsymbol{\theta}) = 0, \exists \boldsymbol{\theta} \in \mathbf{B}_i$ **then**
 9:        **break**
10:      **end if**
11:      $\mathcal{J}_i \leftarrow -\lambda_i \mathcal{L}\big(h(\mathbb{SF}(\mathbf{x}, a_i; \mathcal{M})), h(\mathbf{x})\big) - \sum_{\boldsymbol{\theta}_i \in \mathcal{B}_i} \frac{\lambda_i}{|\mathbf{B}_i|} \mathcal{L}\big(h(\boldsymbol{\theta}_i), h(\mathbf{x})\big) + \hat{P}(\mathbf{x}, a_i)\hat{G}(\mathbf{x}, a_i)$
12:      $a_i \leftarrow \mathrm{Proj}\big(a_i + \tau \nabla_{a_i} \mathcal{J}_i\big)$
13:      $\lambda_i \leftarrow \eta \lambda_i$
14:    **end while**
15:    $\mathbf{R}_{SF} \leftarrow \mathbf{R}_{SF} \cup \{\mathbb{SF}(\mathbf{x}, a_i; \mathcal{M})\}$
16:    $i \leftarrow i + 1$
17:    **if** $i \geq m$ **then**
18:      **break**
19:    **end if**
20: **end for**

---

# F   Individual Dataset Results

The results are presented in Figure 4.

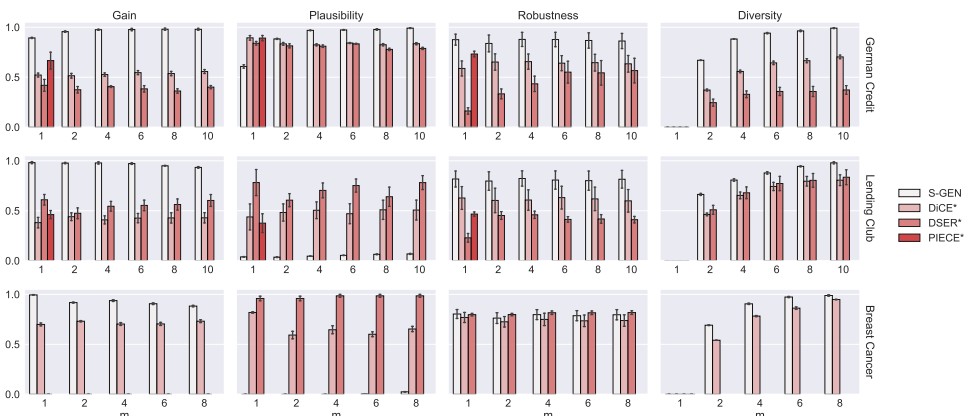

Figure 4: Results: The ability of S-GEN to create semifactuals is compared to DiCE* and PIECE*. Overall, S-GEN does the best, achieving significantly better results to both baselines on 11/16 tests. Moreover, S-GEN was only significantly worse than either baseline on a single test (i.e., plausibility on German Credit), with the remaining four tests being competitive between methods. Standard error bars are shown.

# G Baselines

## G.1 Non-Casual

**DiCE** Our modification to DiCE, starts by generating a counterfactual(s) for a query. Next, we use the algorithm again, but on the generated counterfactuals(s), to make them generate a second counterfactual, which goes back over the decision boundary. In effect, this generates a semifactual(s) for a query.

**PIECE** Second, we use the PIECE framework by Kenny and Keane [29], but apply it to tabular data. Following the authors, we divide the training data into two sets, the first corresponding to those predicted as the original class $c$, and the second to those predicted as the counterfactual class $c'$, these are again split into the respective features. Hence, if there are 2 classes, with 4 features, there are $2 \times 4 = 8$ sets of data. These sets were then modeled using the best fit found for a Beta distribution on continuous features, and a simple Categorical distribution for categorical features. To generate a semifactual predicted as $c$, we take the probability of each feature value in the query using the models of the counterfactual class $c'$, and modify each to be its expected statistical value in $c'$ one-by-one (from the lowest probability to the highest), until the next would take it over the decision boundary. In the case of continuous features, as done by Kenny and Keane [29], we take the probability as being the minimum of the two integrals either side of the feature value in the distribution. In the case the expected feature values lie outside the actionability range, we clip them to the closest value allowed.

**DSER** For Diverse Explanation of Reject [1] (DSER) we had to modify the the technique in two main ways. Most notably, the techniques doesn't deal with categorical features, so to overcome this, we optimised treating all one hot encoded features as real-valued, and then projected each categorical feature onto its nearest value. Next, the method addresses diversity by iterating all different sets of possible features, in our domains this is computationally intractable. Hence, we optimise one semifactual at a time, each time pushing each solution as far as possible from those already found.

## G.2 Causal

**Karimi et al. (2021)** The method by Karimi et al. [25] is a recourse method designed to minimise cost whilst traversing the decision boundary. To modify the technique, we simply stop the optimization when the next step would take it over the decision boundary.

**Dominguez et al. (2022)** The method by Dominguez et al. [18] is identical to Karimi et al. [25], but they add in a robustness component. Namely, they take an individual $x$, and solve an inner loss which means that an individual of distance $\epsilon = 0.1$ (in our tests) close to $x$, with the same recourse given, will also achieve recourse. We simply keep the same optimization process, but aim to solve a different objective. The objective we solve is to move towards the decision boundary, but when the recourse option causes either $x$ or the individual close to it to cross the decision boundary, we terminate the optimization one step prior to this.

# H Computational Costs

All tests were run on a MacBook Pro, Apple M1 Pro, 16 GB. Re-running tests will take less than 1 day.

# I User Study

Here we show our entire user study for complete transparency. We used the German Credit dataset, but converted the currency into U.S. dollars since it was given to U.S. citizens to complete.

**Intro Brief**

Thank you for clicking on this study!

***Do no take this study on a mobile phone, the tables and images wont display correctly.***

Please don't take this study if you did a similar one recently.

 You are free to leave at any time.

 The study will take around 8mins.

 You will be paid $12 per hour for your efforts.

 Thank you for your participation!

**Enter ID**

Please Enter Your Prolific ID Here



**Introduction**

# Introduction

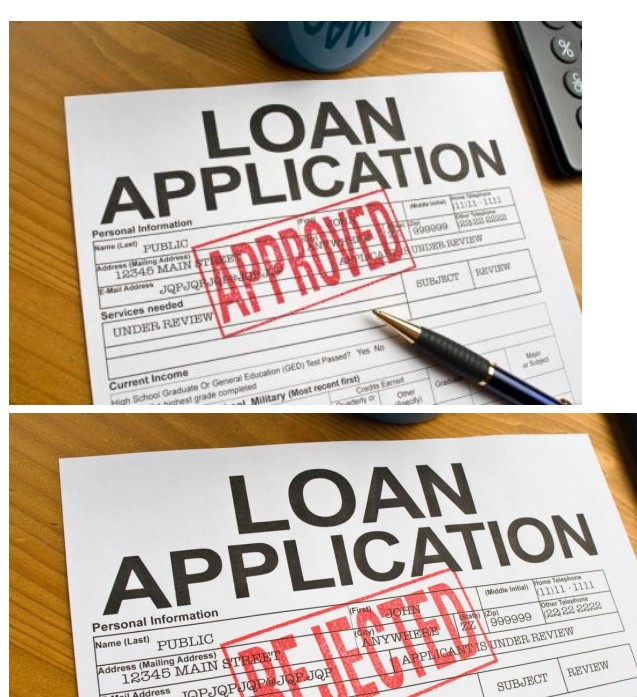

You are going to be shown **six situations** in which a person either has a loan application **approved**, or **rejected**.

You will then be shown **two** different pieces of information for each situation that a bank clerk *could tell* the person.

You are then asked to rate how ***useful*** each of these are. That is, could the information possibly be useful in any way? Or is it not that useful?

Each situation has 4 "features".

# Features Used in Decision

The four "features" used to decide if each person has their loan **approved** or **rejected** are:

**1: Duration**: Over how long the applicant wishes to pay back the loan.

**2: Amount**: How much are they asking to loan from the bank.

**3: Savings**: How much money does the applicant have saved.

**4: Credit Cards**: How many credit cards does the applicant have.

These are the only features the bank clerk uses to make decisions.

**Click Next**

# Now, Please Practice On The Next Question

**Sample Question**

## Example Question

Lucas had his bank loan *accepted*, his features are:

| Duration | 12 Months |
|---|---|
| Amount | $2,000 |
| Savings | $500 |
| Credit Cards | 2 |

The two possible things the bank clerk could tell him are:

**Option 1:** Even if you want to increase your **Duration** to 14 months, and **Amount** to $3,000, we will still accept your loan application.

**Option 2:** If your **Savings** were $100, and your **Credit Cards** 5, we would have rejected your loan application.

How _**useful**_ is each option?

|  | Not Useful |  |  |  | Very Useful |
|---|---|---|---|---|---|
| Option 1 | ○ | ○ | ○ | ○ | ○ |
| Option 2 | ○ | ○ | ○ | ○ | ○ |

**Click Next 2**

# Please only participate in this study if you understand the instructions well

**Block 15**

# Remember, the key question is how _USEFUL_ is each option

**Click Next 3**

# Click Next To Begin The Study

**Question 1**

Kate had her bank loan *accepted*, her features are:

| Duration | 6 Months |
|---|---|
| Amount | $932 |
| Savings | Over $1000 |
| Credit Cards | 2-3 |

The two possible things the bank clerk could tell her are:

**Option 1:** Even if you want to increase your **Amount** to $2,841, and increase your number of **Credit Cards** to 4-5, we will still accept your loan application.

**Option 2:** If your **Duration** was 44 months, and you had had **Savings** less than $100, we would have rejected your loan application.

How *useful* is each option?

|  | Not Useful |  |  |  | Very Useful |
|---|---|---|---|---|---|
| Option 1 | ○ | ○ | ○ | ○ | ○ |
| Option 2 | ○ | ○ | ○ | ○ | ○ |

**Question 2**

Paul had his bank loan *accepted*, his features are:

| Duration | 18 Months |
|---|---|
| Amount | $1,239 |
| Savings | Over $1,000 |
| Credit Cards | 1 |

The two possible things the bank clerk could tell him are:

**Option 1:** Even if you want to increase your **Duration** to 21 Months, and lower your **Savings** to $500-$1,000, we will still accept your loan application.

**Option 2:** If you asked for an **Amount** of $15,499 (or more), and had had 6 **Credit Cards**, we would have rejected your loan application.

How **_useful_** is each option?

|  | Not Useful |  |  |  | Very Useful |
|---|---|---|---|---|---|
| Option 1 | ○ | ○ | ○ | ○ | ○ |
| Option 2 | ○ | ○ | ○ | ○ | ○ |

**Question 3**

Xue had her bank loan **_accepted_**, her features are:

| Duration | 9 Months |
|---|---|
| Amount | $1549 |
| Savings | Over $1,000 |
| Credit Cards | 1 |

The two possible things the bank clerk could tell her are:

**Option 1:** Even if you want to increase your **Duration** to 25 Months, and increase your **Amount** to $4,620, we will still accept your loan application.

**Option 2:** If you had had 3 **Credit Cards**, and no **Savings**, we would have rejected your loan application.

How ***useful*** is each option?

|  | Not Useful | | | | Very Useful |
|---|---|---|---|---|---|
| Option 1 | ◯ | ◯ | ◯ | ◯ | ◯ |
| Option 2 | ◯ | ◯ | ◯ | ◯ | ◯ |

**Question 4**

Siddarth had his bank loan ***rejected***, his features are:

| Duration | 48 Months |
|---|---|
| Amount | $6,143 |
| Savings | None |
| Credit Cards | 2-3 |

The two possible things the bank clerk could tell him are:

**Option 1:** Even if you increase your **Savings** to $100, and lower your number of **Credit Cards** to 1, we will still reject your loan application.

**Option 2:** If you lower your **Duration** to 15 Months, and lower your **Amount** to $4,627, we will accept your loan application.

How ***useful*** is each option?

|         | Not Useful |   |   |   | Very Useful |
|---------|:---:|:---:|:---:|:---:|:---:|
| Option 1 | ○ | ○ | ○ | ○ | ○ |
| Option 2 | ○ | ○ | ○ | ○ | ○ |

**Question 5**

Camila had her bank loan ***rejected***, her features are:

| Duration | 60 Months |
|----------|-----------|
| Amount | $15,653 |
| Savings | None |
| Credit Cards | 2-3 |

The two possible things the bank clerk could tell her are:

**Option 1:** Even if you increase your **Duration** to 70 Months, and reduce your number of **Credit Cards** to 1, we will still reject your loan application.

**Option 2:** If you reduce your **Amount** to $7,296 (or less), and you get $1,000+ **Savings**, we will accept your loan application.

How ***useful*** is each option?

|         | Not Useful |   |   |   | Very Useful |
|---------|:---:|:---:|:---:|:---:|:---:|
| Option 1 | ○ | ○ | ○ | ○ | ○ |
| Option 2 | ○ | ○ | ○ | ○ | ○ |

**Question 6**

Angelo had his bank loan *__rejected__*, his features are:

| | |
|---|---|
| Duration | 60 Months |
| Amount | $7,408 |
| Savings | Less than $100 |
| Credit Cards | 2 |

The two possible things the bank clerk could tell him are:

**Option 1:** Even if you decrease your **Amount** to $6,505, and increase your **Savings** to over $1000, we will still reject your loan application.

**Option 2:** If you lower your **Duration** to 5 Months, and you reduce your number of **Credit Cards** to 1, we will accept your loan application.

How *__useful__* is each option?

|  | Not Useful | | | | Very Useful |
|---|---|---|---|---|---|
| Option 1 | ○ | ○ | ○ | ○ | ○ |
| Option 2 | ○ | ○ | ○ | ○ | ○ |

**Debrief**

# Debrief: You Have Reached The End
Thank you for your participation, this study was designed to evaluate what kind of explanation people prefer from an artificial intelligence system.

