# OpenReview forum: "The Utility of “Even if” Semifactual Explanation to Optimise Positive Outcomes"
_NeurIPS.cc/2023/Conference — NeurIPS 2023 poster_

### Official Review · Reviewer_4usU · 2023-06-23

**Soundness:** 2 fair
**Presentation:** 2 fair
**Contribution:** 3 good
**Rating:** 3
**Confidence:** 3

**Summary:**

This paper proposes a framework for "semi-factual" explanations - explanations given to an applicant who receives an accept decision, about what they could do which would allow them to maintain that decision. The framework and objective propose utilize a causal model of the action space, and consider the applicant's gain in utility, the plausibility of the output, the diversity of suggestions and the robustness of the suggested actions. Experimentally, they demonstrate an improvement of their approach over baselines and also conduct a human study showing their approach is perceived to produce more useful explanations.

**Strengths:**

- incorporation of causal models in counterfactual-type explanations is a good direction and a sensible idea
- the presented framework is quite thorough in terms of the desiderata you'd hope for in semi-factual explanations
- the human study at the end is very cool - I haven't seen so many of these in this literature and I think that really sets this paper apart

**Weaknesses:**

- Overall, I find myself lacking quite a bit of clarity around many of the formalisms in pages 3-6 as the framework is described. Some of this points of confusion are:
1. I find 3.1 hard to follow - for instance, it's not clear to me what kind of object A is: it seems to be from a subset of the domain of X, but then it's stated later that they're not simply added, so I'm not sure how to interpret its value, or what it means that "actions have to be mutable" (line 94); I'm not so sure why we need S if we have S_M, what is S supposed to be?
2. It's not obvious to me what P is supposed to be - should it be conditional on x rather than a marginal? is it defined with reference to some action history? How do we define it in a way which serves us? Also, I'm not sure why P and G should be multiplied - may need more explanation for that
- I found Section 4 challenging to follow as well:
1. it's not clear to me what a "causal" vs a "non-causal" domain is supposed to be
2. I didn't follow the methods for empirical approximations of H or M, and I'm not sure why P needs an approximation if it's just calculating a known density that we define?
3. I don't really follow the fitness function section either but I'm not so familiar with genetic algorithms so that could be why

- I'm not sure that Lemma 3.3 isn't trivial - if H >= 0 for all a, and we assume G > 0 and P, R, \lambda are > 0 by definition, doesn't J have to be >= 0 by definition as well?



Smaller notes:
- how dow we know S inverse exists? (line 105)
- line 120: a little confusing referring to post robustness before it is defined
- line 216: depict is an odd word here, present may be better
- how are the results in Fig 2 normalized? It's not clear to me that they should be
- why is P lowers for S-GEN? Is this good or bad?


**Questions:**

- Is there work from the counterfactual (rather than semi factual) literature that uses causal modelling? I'm not sure how deep the novelty is supposed to run here

**Limitations:**

I recommend against using COMPAS as a dataset in papers which are not in a criminal justice context - see "It's COMPASlicated: The Messy Relationship between RAI Datasets and Algorithmic Fairness Benchmarks" for more, and it's probably worth writing a piece about this.

---

> ### Author Rebuttal · Authors · 2023-08-07
>
> We thank the reviewer for noting the relevance of our user study and how it sets the paper apart. We are truly grateful for your suggestions which have improved the paper, we now respond to your questions & concerns. Apologies for the brevity of some points, but space was limited.
>
> ***
>
> ### Reviewer's Main Question:
>
> Yes there is some work recently, perhaps most relevant is the two works we cited and compared to [1,2].
>
> Our causal algorithm is different to prior work because it uses (1) a maximization objective, and (2) a "Gain" function. Together, these aim to produce the nuances of “even if” thinking we are intending for (which other approaches don’t).
>
> However, we would like the emphasize that this algorithm is not our main contribution, it was instead the conceptual idea to use semi-factuals (and indeed XAI in general) for explaining how to optimize positive outcomes (lines 30-33). Moreover, this involved creating an approach for when the causal structure is not known (i.e., our non-causal evolutionary algorithm), verifying people find the explanations useful (i.e., our user study), and laying a strong theoretical foundation for semi-factual research going forward (i.e., our proofs & derivations).
>
> The causal experiment primarily exists to show how the nuances of “even if” thinking are better captured with causality (lines 313-316), and indeed there was a very strong effect size (lines 322-323), which gives practical validity to Theorem 3.1.
>
> ***
> ### Other Q's
>
> **Object A & Mutable:** A is simply a set of actions the user “does” (e.g., increase their downpayment on a loan). Regarding the addition, we feel that a “+” operator is not the best notation to use here because categorical features are involved. We’d be happy if the reviewer could recommend a better way. By “actions have to be mutable” we mean that the feature should be possible for the user to change, e.g. not “Race”. We have made these points clearer.
>
> **Clarify S and S_M:** S is for the non-causal setting while S_M is for the causal setting. We explicitly mention this now.
>
> **What is P?**  P is a plausibility function, here we realise this with the NN-Dist metric from [3] in non-causal settings, and the SCM in causal ones (line 233). We now clearly note that the SCM guarantees plausibility in casual settings in Sec 4.1 and the “Plausible Gain” paragraph in Sec 3.4
>
> **Should P be conditional on x?**  Correct, that is actually how we implemented P (line 257).
>
> **Is x defined with action history:** No, we left sequential decision making for future work (see line 71).
>
> **How to define P to serve us?** This is an open question, prior work has shown the NN-Dist metric works well [3], but the advantage of SCMs is that we don’t have to worry about it.
>
> **Why P & G multiplied?** We regard P as a scaling factor for G, together it can be seen as an expected gain where a high plausibility ensures high expected gain. Likewise, 0 plausibility ensures that we will have a 0 expected gain. You can add them, but doing so misses this special property. We mention this now.
>
> **Explain causal" v. non-causal:** The causal setting here is when the SCM and structural equations are known, whilst the latter is when you ignore causal dependencies.
>
> **How approximate H?** It has two parts: Firstly, $\hat{H}_p$ refers to the robust $e$-neighborhood around a semi-factual that is approximated with MC sampling. Secondly, $\hat{H}_a$ is the robustness of the semi-factual itself being classified correctly. We made this clearer in Sec 4.
>
> **How approximate M?** We assume the reviewer is using M to refer to the causal model? We were not approximating it as it is pre-defined.
>
> **Why P need approximation?** Because the distribution of most real world data is unknown. Line 181 is just a theoretical formulation, in practice we need to estimate it, for us this is the NN-Dist metric from [3], whilst the SCMs guarantee it in casual situations. We now note this about P in Sec 3.4.
>
> **Fitness Function Clarity:**  We included an additional sentence in Sec 4.2. explaining fitness function background.
>
> **Lemma 3.3 Trivial?** The main focus of Lemma 3.3 is to verify that our simplification from Eq. (8) to formulation Eq. (9) is correct. As an aside, we also want to show that if the constraints are satisfied, the adversarial objective will just go to infinity. Another way to think about it is the score becomes reasonable when the constraints are all satisfied, and in our task this means that robustness must be satisfied to get reasonable Gain.
>
> **S inverse exists?** S inverse does not always exist and this is usually decided by the specific dataset where a causal graph exists. It is not a rigorous function but can simply be some rule-based mapping. It follows the same flow of thought in related work like [1].
>
> **Post robustness & line 216:** fixed both.
>
> **Fig 2 normalization:** We used min-max normalisation. We did it because all datasets have different ranges in their results, so we can’t just average them normally. We now note this in Fig 2 and added the non-normalized plots in the Appendix. We also uploaded this as a PDF in the global response above for the reviewer's convenience.
>
> **Why P lower for S-GEN?** In our non-causal tests lower plausibility is better, so yes S-GEN performs best in Fig 2. Plausibility here is the distance of the generated semi-factual to the nearest training datapoint, we borrowed the metric from [3]. We now note this in the Fig caption.
>
> **COMPAS:** Thanks. We used it because of the availability of high quality SCMs and structural equations, as is common practice [1]. In the future we will avoid using this dataset.
>
> ***
>
> [1] Dominguez-Olmedo, et al. "On the adversarial robustness of causal algorithmic recourse." ICML
>
> [2] Karimi et al. "Algorithmic recourse: from counterfactual explanations to interventions." FAccT.
>
> [3] Kenny & Keane. "On generating plausible counterfactual and semi-factual explanations for deep learning." AAAI

---

> > ### Comment · Reviewer_4usU · 2023-08-11
> > **Response**
> >
> > Thanks for your rebuttal - I'll respond point by point here:
> >
> > Novelty: to clarify, is the novelty here in the optimization framework for semi-factuals? How does this differ from the image-based works in the beginning of Sec 2 - do they not contain optimization frameworks as well?
> >
> > Object A: I'm still not so clear - in general I understand we might want some more complex operation to combine A with X to produce a new point, but I don't see then how A can be from the domain of X. Additionally, I don't understand how "race" could be a potential action, isn't that a feature? (mutability aside)
> >
> > S and S_M: I'm still not sure I understand - they both seem to be from X* x A* -> X*, defining semi-factual interactions. Don't they both define the same function - if S_M is associated with an SCM, wouldn't it all be associated with S?
> >
> > P: what is a plausibility function? It seems like there's quite a bit around P which needs to be added to the paper for clarity's sake.
> >
> > P conditional on X: I meant, should it be conditional on the initial value of X (rather than the full dataset as you've defined it)
> >
> > Causal vs non-causal: I guess this is nit-picking but I would find it clearer if you referred to these as two "modelling approaches" rather than "domains" - whether or not you happen to know an SCM is not a feature of the domain
> >
> >
> > In general, it's hard for me to re-evaluate clarity without seeing the draft. It seems that at the moment we still have a few major points where I'm not totally following - for this reason I'm inclined to not change my score. However, I see that I'm the odd one out here among reviewers  so it's possible that I'm missing background within this literature, so I will consider this as I discuss with them.

---

> > > ### Author Response · Authors · 2023-08-13
> > > **Author(s) Second Response (Part 1)**
> > >
> > > We appreciate the follow-up questions and do our best to clarify all residual concerns. Note to respond adequately, we have divided this response into two parts (this is the first).
> > >
> > > ***
> > >
> > > *Novelty: to clarify, is the novelty here in the optimization framework for semi-factuals? How does this differ from the image-based works in the beginning of Sec 2 - do they not contain optimization frameworks as well?*
> > >
> > > Correct, our novelty is primarily with our optimization framework for semi-factuals. However, we do also consider our user study a major contribution (see lines 40-41), as e.g. [3] recently discussed “the paucity of user studies” as a major pitfall in current semi-factual research (indeed the reviewer also graciously noted this sets our paper apart).
> > >
> > > Yes, the image-based works in Section 2 do contain semi-factual optimization frameworks also, however, to answer your question ours has two important differences:
> > >
> > > Firstly, prior work typically defines a “good” semi-factual just by how large the $L_2$ distance is between the test instance and the generated semi-factual explanation [2]. In contrast, we define good semi-factuals as those with high “Gain” scores, which is designed to be more meaningful in application by purposefully aiming to change features in a way that would benefit a user and correspond with “even if” thinking (see lines 126-137).
> > >
> > > Secondly, our framework “flips” the usual recourse setup to focus on *optimizing positive outcomes*, rather than *changing negative outcomes into positive ones*, to the best of our knowledge, this is a novel problem formulation for explainable AI.
> > >
> > > ***
> > >
> > > *Object A: I'm still not so clear - in general I understand we might want some more complex operation to combine A with X to produce a new point, but I don't see then how A can be from the domain of X. Additionally, I don't understand how "race" could be a potential action, isn't that a feature? (mutability aside)*
> > >
> > > A is not from the domain of X but from an action space parameterized with X (defined in line 90-91). It just means that the action space for each feature depends on the feature itself. For example, regarding a feature “employment”, the action space could be {become employed, become unemployed, become self-employed}, which depends on the characteristics of the specific feature, this is all we mean.
> > >
> > > Sorry there appears to have been confusion partly because of the space constraints prior. You are correct that “Race” is a feature and not an action. To be clearer and expand our original rebuttal we can say:  *By “actions have to be mutable” we mean that the feature should be possible for the user to change, e.g. not “doing the action” of changing a “Race” feature from Race A => Race B…”*
> > >
> > >
> > > ***
> > >
> > > *S and S_M: I'm still not sure I understand - they both seem to be from X * x A * -> X * , defining semi-factual interactions. Don't they both define the same function - if S_M is associated with an SCM, wouldn't it all be associated with S?*
> > >
> > > No, they do not define the same function. S_M associates with a causal model M, whilst S does not consider the causal model. That is, with S, the only changed features correspond to the actions of x. However, in S_M, other features with dependence on the action-changed features might also change accordingly.
> > >
> > > For example, a user might decide to do the action of lowering their income, in S this is the only feature that would change, but in S_M, it would also naturally change the “dept to income ratio” feature due to the causal dependencies.

---

> > > > ### Author Response · Authors · 2023-08-13
> > > > **Author(s) Second Response (Part 2)**
> > > >
> > > > We appreciate the follow-up questions and do our best to clarify all residual concerns. Note to respond adequately, we have divided this response into two parts (this is the second).
> > > >
> > > > ***
> > > >
> > > > *P: what is a plausibility function? It seems like there's quite a bit around P which needs to be added to the paper for clarity's sake.*
> > > >
> > > > A plausibility function as we define it here is simply a function which outputs a value corresponding to how “plausible” a semi-factual explanation is. The plausibility of a generated explanation is extremely well studied in the literature (e.g., see [1,2,4] for just some examples).
> > > >
> > > > In our case we simply use the distance of generated semi-factuals to known training data to do this here in non-causal settings (see lines 254-259; similar to [2]). But again, plausibility is naturally taken care of (at least how we define it here) thanks to the SCM in causal settings.
> > > >
> > > > For what it’s worth, we don’t feel this will be difficult to articulate further in the paper to be crystal clear. Indeed, one simple sentence at the end of our “Plausible Gain” paragraph on page 5 should do it. We would have included this in our initial response to the reviewer but had no space left, so we can thankfully do it here:
> > > >
> > > > **Revision:** We add this at the end of the “Plausible Gain” paragraph on page 5: ” *“Note however, in our non-causal tests we use the $L_2$ distance to the actual training data to approximate plausibility (i.e., being in distribution, similar to [1,2,4]), but this issue of plausibility is naturally taken care of in our causal tests thanks to the SCM ensuring plausible feature mutations, so we don’t explicitly consider plausibility there going forward.*
> > > >
> > > > ***
> > > >
> > > > *P conditional on X: I meant, should it be conditional on the initial value of X (rather than the full dataset as you've defined it)*
> > > >
> > > > Thanks for clarifying this, and sorry for the misunderstanding. Indeed P could be conditional on X only. However, in practice, it is hard to get the true P, and we thus appealed to an empirical approximation (line 257) as most work in this area does (see [1,2,4]). In our setting, the empirical approximation is built using the training dataset directly.
> > > >
> > > > ***
> > > >
> > > > *Causal vs non-causal: I guess this is nit-picking but I would find it clearer if you referred to these as two "modelling approaches" rather than "domains" - whether or not you happen to know an SCM is not a feature of the domain*
> > > >
> > > > No problem, done.
> > > >
> > > > ***
> > > >
> > > > *In general, it's hard for me to re-evaluate clarity without seeing the draft. It seems that at the moment we still have a few major points where I'm not totally following - for this reason I'm inclined to not change my score. However, I see that I'm the odd one out here among reviewers so it's possible that I'm missing background within this literature, so I will consider this as I discuss with them.*
> > > >
> > > > We hope this has clarified the points above which remained unclear, if not please let us know and we are happy to do more, thanks again.
> > > >
> > > > ***
> > > >
> > > > [1] Laugel, Thibault, et al. "The dangers of post-hoc interpretability: unjustified counterfactual explanations." IJCAI. 2019.
> > > >
> > > > [2] Kenny, Eoin M., and Mark T. Keane. "On generating plausible counterfactual and semi-factual explanations for deep learning." AAAI
> > > >
> > > > [3] Aryal, Saugat, and Mark T. Keane. "Even if explanations: Prior work, desiderata & benchmarks for semi-factual XAI." IJCAI – 2023
> > > >
> > > > [4] Van Looveren, Arnaud, and Janis Klaise. "Interpretable Counterfactual Explanations Guided by Prototypes."

---

### Official Review · Reviewer_vFfC · 2023-07-06

**Soundness:** 3 good
**Presentation:** 3 good
**Contribution:** 2 fair
**Rating:** 7
**Confidence:** 3

**Summary:**

This paper looks into the mirror problem of counterfactual recourse. In the latter the goal is to determine what is needed to change a negative outcome to a positive such as in loans. In the former the goal is to determine to what extend a positive decision would remain and to what extend the end user could have pushed their features. The paper introduces this novel view with motivations from psychology and also investigates this with a user study. In order to achieve this they use similar ideas from causal algorithm recourse with constraints on the proposed explanations.

**Strengths:**

- The paper tackles a new problem that has not had much attention in the XAI community. We are often focused on the algorithm recourse, however, this paper proposes a new twist to explanations when users in fact got a positive outcome. The user study albeit only 15 samples (might be wrong on this) shows that people do in fact prefer this type of explanation when in fact getting a positive outcome. Which tbh makes sense as in why would they care about what would have happened if they didn't get it?
- Another plus is the simplicity of the idea which i like very much given that it is well-motivated.
- The paper also clearly introduces their desiderata for the explanations as well as the objective they aim to solve.
- Lastly, just like in the previous causal algorithm recourse, they also provide a causal view of this problem.

**Weaknesses:**

- The paper seems to take too much inspiration from the causal algorithm recourse paper to a point where it becomes just a rewriting of that paper from a different view with the exact same/similar techniques.
- I also don't understand how gain and cost are not equivalent as you can have causal costs as well. in your line 145 you say the main difference is the causal part which I don't get as this can also be done with cost no?
- In Figure 2, what happened to plausibility? does that mean the explanations are all non-plausible?
-

**Questions:**

see above

---

> ### Author Rebuttal · Authors · 2023-08-07
>
> We sincerely thank the reviewer for noting the novelty and simplicity (in a good way) of our idea and its relevance to the field. We now address all questions and concerns:
>
> ***
>
> *The paper seems to take too much inspiration from the causal algorithm recourse paper to a point where it becomes just a rewriting of that paper from a different view with the exact same/similar techniques.*
>
> Our causal algorithm is different to prior work because it uses (1) a maximization objective, and (2) a "Gain" function. Together, these aim to produce the nuances of “even if” thinking we are intending for (which other approaches don’t).
>
> However, we would like the emphasize that this algorithm is not our main contribution, it was instead (see line 38) the conceptual idea to use semi-factuals (and indeed XAI in general) for explaining how to optimize positive outcomes (see lines 30-33). Moreover, this involved creating an approach for when the causal structure is not known (i.e., our non-causal algorithm), verifying people find the explanations useful (i.e., our user study), and laying a strong theoretical foundation for semi-factual research going forward (i.e., our proofs and derivations).
>
> The causal experiment primarily exists to show how the nuances of “even if” thinking are better captured with causality (see lines 313-316), and indeed there was a very strong effect size (lines 322-323), which gives practical validity to our proof in Theorem 3.1.
>
> ***
>
> *I also don't understand how gain and cost are not equivalent as you can have causal costs as well. in your line 145 you say the main difference is the causal part which I don't get as this can also be done with cost no?*
>
> Apologies for the lack of clarity, our gain function requires a set of human defined directions for features to mutate in order to be considered positive/negative gain (e.g., if a user desires to work less hours per week then lowering this feature = positive gain, and raising it = negative gain; see lines 131-133), which cost does not consider, this alone means it can be different (see line 136).
>
> To answer your question however, our experience is that cost is usually formulated as the actions an end user takes to achieve recourse [1,2]. This makes sense since even if other features change causally, it didn’t “cost” the user anything to modify them in terms of effort. However, gain doesn’t only care about the “effort” taken by the user, but also how much they stand to gain from the resultant causal mutation, so ideally, gain will *always* consider causality, whilst cost doesn't necessarily (although we understand that isn't always practical, so we included the non-causal algorithm too).
>
> **Revision:** We emphasise how a Gain function (in contrast to cost) requires the specification of positive/negative directions for each feature in our “Why is gain not necessarily equivalent to cost?” paragraph.
>
> ***
>
> *In Figure 2, what happened to plausibility? does that mean the explanations are all non-plausible?*
>
> Apologies but we assume the reviewer is referring to Figure 3? Plausibility is shown in Figure 2. In this work we define plausibility as the generation of explanations which are within distribution. Thanks to the SCM, this is guaranteed in the causal experiments, so we don’t measure plausibility there.
>
> **Revision:** We made this clear at the end of Section 5 by adding *“Note for causal tests the SCM guarantees plausibility so this metric is not reported”*.
>
> ***
>
> [1] Karimi, Amir-Hossein, et al. "A survey of algorithmic recourse: definitions, formulations, solutions, and prospects."
>
> [2] Dominguez-Olmedo, Ricardo, Amir H. Karimi, and Bernhard Schölkopf. "On the adversarial robustness of causal algorithmic recourse." International Conference on Machine Learning. PMLR, 2022..

---

> > ### Comment · Reviewer_vFfC · 2023-08-18
> > **Thanks for the clarification**
> >
> > I have read the reply and would like to thank the reviewers for the clarifications.
> >
> > I have an additional question however, you state that your main contribution is not the optimization but rather the introduction of the concept. Hence, could the authors please tell me exactly what the differences to are "On generating plausible counterfactual and semi-factual explanations for deep learning." besides that they are applying it to images.
> > Please be specific and clearly state your novelty besides the new data type setting.
> > I would also like to note that the paper has been cited 74 times and hence urge the authors to be specific on the related works. I will myself look through the citations and cross check your answer later.
> >
> > Thanks

---

> > > ### Author Response · Authors · 2023-08-21
> > > **Author(s) response to additional question**
> > >
> > > We thank the reviewer for their engagement with our work. We start by first addressing how our research differs from Kenny & Keane [1], and then the 74 citations.
> > >
> > > ### Difference to [1]
> > > Our novel “concept” is to optimise positive outcomes, in [1] and all 74 citations, no one else has considered this. This is important because despite their growing popularity, it is not known how semi-factuals (SFs) could be useful for ML. Indeed [5] discuss this issue explicitly *“it is… unclear… how these types of explanations are useful…”* [5]. So, to rectify this, we showed in our user study that optimising positive outcomes with SFs is useful.
> > >
> > > To actualise this, we made many advances to [1].
> > >
> > > |                    | Kenny & Keane [1] | Ours                                                                             |
> > > |--------------------|-------------------|----------------------------------------------------------------------------------|
> > > | Framework| Modify features of low probability in the counterfactual (CF) class to be their expected values. Stop this one step before the decision boundary to generate SF. | Maximise the “distance” between a test instance and its SF using an SF specific optimisation with a gain function, actionability constraints, and SCMs. |
> > > | Theoretical Contribution | None| Yes, verification of the SF objective and comparison to CFs.|
> > > | User Study| None| Yes, showed a use case in which SFs are more useful compared to CFs (something missing from the XAI lit. but desired [5,7]) |
> > > | Causality Considered |No| Yes, full derivations, objectives, theory, and experimental results.|
> > > | Categorial features considered| No|Yes|
> > > | Problem Setting| Deep learning – image classification | Algorithmic Recourse|
> > >
> > > Note the distinction between images and tabular is important in recourse [2], and we outperformed Kenny & Keane [1] significantly in comparative tests.
> > >
> > > ### 74 Citations
> > > Of these, 5 propose a new SF algorithm, 4 are not directly related to XAI, 1 uses SFs for evaluation, 9 are surveys, 29 cite in relation to CFs, 4 are related to CBR, 4 discuss user testing (but don’t do it), 8 discuss SFs, 1 is from the psychology, and 9 we classify as “Other” (e.g. a thesis). We now detail the 5 SF algorithms (we didn’t cite [3,4] in our paper since they are not published):
> > >
> > > [3] use a joint gaussian mixture model to sample CF (and SF) images in a deep learner. They don’t consider individual features, categorical features, recourse, users, causality, gain, or positive outcomes. (not published)
> > >
> > > [4] was released on arXiv the same month as the NeurIPS deadline, so as per NeurIPS guidelines we do not consider this. (not published)
> > >
> > > [6] and [8] both use standard CF approaches with generative models on images similar to [1], and stop before the decision boundary to make SFs. Both (and [1,3]) see SFs as an "afterthought" to CFs, ignore users, categorical features, theory, recourse, and positive outcomes.
> > >
> > > The most relevant work is [5], as they propose an SF specific algorithm. They suggested using diverse SFs to explain reject decisions. The authors note that their method (1) likely produces implausible explanations, and (2) that they are not clearly useful, partly because how to formalise SFs is unclear. Here, we address this: (1) We guarantee plausibility with SCMs and actionability constraints, and (2) we showed people find our problem formulation and explanations useful with our user study.
> > >
> > > ### Conclusion
> > > We trust the reviewer can see how our work differs and advances SF research. Above all, our concept of using SFs to optimise positive outcomes has compelling evidence (in our user testing) that it is **perhaps the first useful formulation for SFs in ML**, which is a big step forward for SF research
> > >
> > > So, to be clear, aside from the framework of optimising positive outcomes, our Gain functions, theory, and user study are completely novel compared to all prior work.
> > >
> > > Thanks
> > >
> > > P.S. We post all cites and our categories of them below so the reviewer may more easily fact check us.
> > >
> > > ***
> > >
> > > [1] Kenny, Eoin M., and Mark T. Keane. "On generating plausible counterfactual and semi-factual explanations for deep learning." AAAI
> > > [2] Verma, Sahil, et al. "Counterfactual explanations and algorithmic recourses for machine learning: A review." arXiv
> > >
> > > [3] Xie, Zhouyang, and Duanbing Chen. "Joint Gaussian Mixture Model for Versatile Deep Visual Model Explanation."
> > >
> > > [4] Dandl, Susanne, et al. "Interpretable Regional Descriptors: Hyperbox-Based Local Explanations." arXiv
> > >
> > > [5] Artelt, André, and Barbara Hammer. "“Even if…”–Diverse Semifactual Explanations of Reject."  IEEE.
> > >
> > > [6] Vats, Anuja, et al. "This changes to that: Combining causal and non-causal explanations to generate disease progression in capsule endoscopy." ICASSP
> > >
> > > [7] Aryal, Saugat, and Mark T. Keane. "Even if explanations: Prior work, desiderata & benchmarks for semi-factual XAI." IJCAI
> > >
> > > [8] Zhao, Ziwei, et al. "Generating Counterfactual Images: Towards a C2C-VAE Approach."

---

> > > > ### Author Response · Authors · 2023-08-21
> > > > **Our Categorisations of Kenny & Keane's Citations (part 1)**
> > > >
> > > > ## Other SF Algorithms
> > > > Xie, Zhouyang, and Duanbing Chen. "Joint Gaussian Mixture Model for Versatile Deep Visual Model Explanation." (2022).
> > > >
> > > > Dandl, Susanne, et al. "Interpretable Regional Descriptors: Hyperbox-Based Local Explanations." arXiv preprint arXiv:2305.02780 (2023). [5] Artelt, André, and Barbara Hammer. "“Even if…”–Diverse Semifactual Explanations of Reject." 2022 IEEE Symposium Series on Computational Intelligence (SSCI). IEEE, 2022.
> > > >
> > > > Artelt, André, and Barbara Hammer. "“Even if…”–Diverse Semifactual Explanations of Reject." 2022 IEEE Symposium Series on Computational Intelligence (SSCI). IEEE, 2022.
> > > >
> > > > Vats, Anuja, et al. "This changes to that: Combining causal and non-causal explanations to generate disease progression in capsule endoscopy." ICASSP 2023-2023 IEEE International Conference on Acoustics, Speech and Signal Processing (ICASSP). IEEE, 2023.
> > > >
> > > > Zhao, Ziwei, et al. "Generating Counterfactual Images: Towards a C2C-VAE Approach." (2022).
> > > >
> > > > ## Not directly related to XAI
> > > > Lu, Jinghui, et al. "A rationale-centric framework for human-in-the-loop machine learning." arXiv preprint arXiv:2203.12918 (2022).
> > > >
> > > > Hagos, Misgina Tsighe, Kathleen M. Curran, and Brian Mac Namee. "Identifying Spurious Correlations and Correcting them with an Explanation-based Learning." arXiv preprint arXiv:2211.08285 (2022).
> > > >
> > > > Ye, Xiaomeng, et al. "Applying the case difference heuristic to learn adaptations from deep network features." arXiv preprint arXiv:2107.07095 (2021).
> > > >
> > > > Leofante, Francesco, and Alessio Lomuscio. "Towards Robust Contrastive Explanations for Human-Neural Multi-agent Systems." Proceedings of the 2023 International Conference on Autonomous Agents and Multiagent Systems. 2023.
> > > >
> > > > ## Using for Evaluation
> > > > Wilkerson, Zachary, David Leake, and David Crandall. "Leveraging SHAP and CBR for Dimensionality Reduction on the Psychology Prediction Dataset." (2022).
> > > >
> > > > ## Survey Papers
> > > > Bodria, Francesco, et al. "Benchmarking and survey of explanation methods for black box models." Data Mining and Knowledge Discovery (2023): 1-60.
> > > >
> > > > Nimmy, Sonia Farhana, et al. "Explainability in supply chain operational risk management: A systematic literature review." Knowledge-Based Systems 235 (2022): 107587.
> > > >
> > > > Keane, Mark T., et al. "Twin systems for deepcbr: A menagerie of deep learning and case-based reasoning pairings for explanation and data augmentation." arXiv preprint arXiv:2104.14461 (2021).
> > > >
> > > > Kenny, Eoin M., et al. "Post-hoc explanation options for XAI in deep learning: The Insight centre for data analytics perspective." Pattern Recognition. ICPR International Workshops and Challenges: Virtual Event, January 10–15, 2021, Proceedings, Part III. Springer International Publishing, 2021.
> > > >
> > > > Poché, Antonin, Lucas Hervier, and Mohamed-Chafik Bakkay. "Natural Example-Based Explainability: a Survey." (2023).
> > > >
> > > > Holmberg, Lars, Paul Davidsson, and Per Linde. "Mapping Knowledge Representations to Concepts: A Review and New Perspectives." arXiv preprint arXiv:2301.00189 (2022).
> > > >
> > > > Nazir, Sajid, Diane M. Dickson, and Muhammad Usman Akram. "Survey of explainable artificial intelligence techniques for biomedical imaging with deep neural networks." Computers in Biology and Medicine (2023): 106668.
> > > >
> > > > Keane, Mark T., et al. "If only we had better counterfactual explanations: Five key deficits to rectify in the evaluation of counterfactual xai techniques." arXiv preprint arXiv:2103.01035 (2021).
> > > >
> > > > Verma, Sahil, et al. "Counterfactual explanations and algorithmic recourses for machine learning: A review." arXiv preprint arXiv:2010.10596 (2020). – This is a great example of how tablar data is a REALLY important distinction
> > > >
> > > > ## Citing in Context of Counterfactuals
> > > > Wang, Zijie J., et al. "GAM Coach: Towards Interactive and User-centered Algorithmic Recourse." Proceedings of the 2023 CHI Conference on Human Factors in Computing Systems. 2023.
> > > >
> > > > Aguilera-Ventura, Carlos, et al. "Counterfactual Reasoning via Grounded Distance." Proceedings of the International Conference on Principles of Knowledge Representation and Reasoning. Vol. 19. No. 1. 2023.
> > > >
> > > > Wang, Chong, et al. "Learning Support and Trivial Prototypes for Interpretable Image Classification." arXiv preprint arXiv:2301.04011 (2023).
> > > >
> > > > Booth, Serena, et al. "Bayes-trex: a bayesian sampling approach to model transparency by example." Proceedings of the AAAI Conference on Artificial Intelligence. Vol. 35. No. 13. 2021.
> > > >
> > > > Höltgen, Benedikt, et al. "Deduce: generating counterfactual explanations at scale." eXplainable AI approaches for debugging and diagnosis.. 2021.
> > > >
> > > > Yang, Linyi, et al. "Exploring the efficacy of automatically generated counterfactuals for sentiment analysis." arXiv preprint arXiv:2106.15231 (2021).

---

> > > > > ### Author Response · Authors · 2023-08-21
> > > > > **Our Categorisations of Kenny & Keane's Citations (part 2)**
> > > > >
> > > > > ...
> > > > >
> > > > >
> > > > > Höltgen, Benedikt, et al. "Deduce: Generating counterfactual explanations efficiently." arXiv preprint arXiv:2111.15639 (2021).
> > > > >
> > > > > Schemmer, Max, et al. "Towards Meaningful Anomaly Detection: The Effect of Counterfactual Explanations on the Investigation of Anomalies in Multivariate Time Series." arXiv preprint arXiv:2302.03302 (2023).
> > > > >
> > > > > Guidotti, Riccardo. "Counterfactual explanations and how to find them: literature review and benchmarking." Data Mining and Knowledge Discovery (2022): 1-55.
> > > > >
> > > > > Yang, Linyi, et al. "Generating plausible counterfactual explanations for deep transformers in financial text classification." arXiv preprint arXiv:2010.12512 (2020).
> > > > >
> > > > > Delaney, Eoin, Derek Greene, and Mark T. Keane. "Uncertainty estimation and out-of-distribution detection for counterfactual explanations: Pitfalls and solutions." arXiv preprint arXiv:2107.09734 (2021).
> > > > >
> > > > > Cho, Soo Hyun, and Kyung-shik Shin. "Feature-Weighted Counterfactual-Based Explanation for Bankruptcy Prediction." Expert Systems with Applications 216 (2023): 119390.
> > > > >
> > > > > Huang, Qinhua, and Weimin Ouyang. "Improving Causality Explanation of Judge-View Generation Based on Counterfactual." International Conference on Intelligent Computing. Singapore: Springer Nature Singapore, 2023.
> > > > >
> > > > > Warren, Greta, Barry Smyth, and Mark T. Keane. "“Better” Counterfactuals, Ones People Can Understand: Psychologically-Plausible Case-Based Counterfactuals Using Categorical Features for Explainable AI (XAI)." International conference on case-based reasoning. Cham: Springer International Publishing, 2022.
> > > > >
> > > > > Albini, Emanuele, et al. "On the Connection between Game-Theoretic Feature Attributions and Counterfactual Explanations." arXiv preprint arXiv:2307.06941 (2023).
> > > > >
> > > > > Jeanneret, Guillaume, Loïc Simon, and Frédéric Jurie. "Diffusion models for counterfactual explanations." Proceedings of the Asian Conference on Computer Vision. 2022.
> > > > >
> > > > > Delaney, Eoin, et al. "Counterfactual explanations for misclassified images: How human and machine explanations differ." arXiv preprint arXiv:2212.08733 (2022).
> > > > >
> > > > > Warren, Greta, et al. "Explaining Groups of Instances Counterfactually for XAI: A Use Case, Algorithm and User Study for Group-Counterfactuals." arXiv preprint arXiv:2303.09297 (2023).
> > > > >
> > > > > Pawelczyk, Martin, Lea Tiyavorabun, and Gjergji Kasneci. "Decomposing Counterfactual Explanations for Consequential Decision xMaking." arXiv preprint arXiv:2211.02151 (2022).
> > > > >
> > > > > Jiang, Junqi, et al. "Formalising the robustness of counterfactual explanations for neural networks." Proceedings of the AAAI Conference on Artificial Intelligence. Vol. 37. No. 12. 2023.
> > > > >
> > > > > Warren, Greta, Ruth MJ Byrne, and Mark T. Keane. "Categorical and continuous features in counterfactual explanations of AI systems." Proceedings of the 28th International Conference on Intelligent User Interfaces. 2023.
> > > > >
> > > > > Zhou, Yilun. "Iterative Partial Fulfillment of Counterfactual Explanations: Benefits and Risks." arXiv preprint arXiv:2303.11111 (2023).
> > > > >
> > > > > Fahse, Tobias Benjamin, Ivo Blohm, and Benjamin van Giffen. "Effectiveness of Example-Based Explanations to Improve Human Decision Quality in Machine Learning Forecasting Systems." (2022).
> > > > >
> > > > > Labaien Soto, Jokin, Ekhi Zugasti Uriguen, and Xabier De Carlos Garcia. "Real-Time, Model-Agnostic and User-Driven Counterfactual Explanations Using Autoencoders." Applied Sciences 13.5 (2023): 2912.
> > > > >
> > > > > Delaney, Eoin, Derek Greene, and Mark T. Keane. "Instance-based counterfactual explanations for time series classification." International Conference on Case-Based Reasoning. Cham: Springer International Publishing, 2021.
> > > > >
> > > > > Na, Seung-Hyup, Woo-Jeoung Nam, and Seong-Whan Lee. "Toward practical and plausible counterfactual explanation through latent adjustment in disentangled space." Expert Systems with Applications 233 (2023): 120982.
> > > > >
> > > > > Patil, Anish, et al. "Utilization of GAN for Automatic Evaluation of Counterfactuals: Challenges and Opportunities."
> > > > >
> > > > > Pawelczyk, Martin, et al. "Carla: a python library to benchmark algorithmic recourse and counterfactual explanation algorithms." arXiv preprint arXiv:2108.00783 (2021).
> > > > >
> > > > > Salazar, Sebastian, Samuel Denton, and Ansaf Salleb-Aouissi. "Counterfactual Explanations for Support Vector Machine Models." arXiv preprint arXiv:2212.07432 (2022).
> > > > >
> > > > > ## Cited in Context of CBR
> > > > > Horta, Vitor AC, and Alessandra Mileo. "Generating Local Textual Explanations for CNNs: A Semantic Approach Based on Knowledge Graphs." International Conference of the Italian Association for Artificial Intelligence. Cham: Springer International Publishing, 2021.
> > > > >
> > > > > Ye, Xiaomeng, et al. "Learning adaptations for case-based classification: A neural network approach." Case-Based Reasoning Research and Development: 29th International Conference, ICCBR 2021, Salamanca, Spain, September 13–16, 2021, Proceedings 29. Springer International Publishing, 2021.

---

> > > > > > ### Author Response · Authors · 2023-08-21
> > > > > > **Our Categorisations of Kenny & Keane's Citations (part 3)**
> > > > > >
> > > > > > ...
> > > > > >
> > > > > > Du, Haiwen, et al. "Can We Transfer Noise Patterns? An Multi-environment Spectrum Analysis Model Using Generated Cases." arXiv preprint arXiv:2308.01138 (2023).
> > > > > >
> > > > > > Leake, David, et al. "Enhancing Case-Based Reasoning with Neural Networks." Compendium of Neurosymbolic Artificial Intelligence. IOS Press, 2023. 387-409.
> > > > > >
> > > > > > ## Positing Use for Semi-factuals in User Testing
> > > > > > Salimi, Pedram. "Addressing Trust and Mutability Issues in XAI utilising Case Based Reasoning." ICCBR Doctoral Consortium 2022 1613 (2022): 0073.
> > > > > >
> > > > > > Mueller, Shane, et al. "Authoring Guide for Cognitive Tutorials for Artificial Intelligence: Purposes and Methods." (2021).
> > > > > >
> > > > > > Hoffman, Robert R., et al. "Evaluating machine-generated explanations: a “Scorecard” method for XAI measurement science." Frontiers in Computer Science 5 (2023): 1114806.
> > > > > >
> > > > > > Mueller, Shane T., et al. "A Computational Cognitive Model of Informative and Persuasive Explanations of Artificial Intelligence Systems."
> > > > > >
> > > > > > ## Discuss Semi-Factuals
> > > > > > Montenegro, Helena, et al. "Privacy-preserving case-based explanations: enabling visual interpretability by protecting privacy." IEEE Access 10 (2022): 28333-28347.
> > > > > >
> > > > > > Kenny, Eoin M., and Mark T. Keane. "Explaining Deep Learning using examples: Optimal feature weighting methods for twin systems using post-hoc, explanation-by-example in XAI." Knowledge-Based Systems 233 (2021): 107530.
> > > > > >
> > > > > > Montenegro, Helena, Wilson Silva, and Jaime S. Cardoso. "Privacy-preserving generative adversarial network for case-based explainability in medical image analysis." IEEE Access 9 (2021): 148037-148047.
> > > > > >
> > > > > > Kenny, Eoin M., et al. "Explaining black-box classifiers using post-hoc explanations-by-example: The effect of explanations and error-rates in XAI user studies." Artificial Intelligence 294 (2021): 103459.
> > > > > >
> > > > > > Warren, Greta, Mark T. Keane, and Ruth MJ Byrne. "Features of Explainability: How users understand counterfactual and causal explanations for categorical and continuous features in XAI." arXiv preprint arXiv:2204.10152 (2022).
> > > > > >
> > > > > > Carraro, Diego, and Kenneth N. Brown. "CouRGe: Counterfactual Reviews Generator for Sentiment Analysis." Irish Conference on Artificial Intelligence and Cognitive Science. Cham: Springer Nature Switzerland, 2022.
> > > > > >
> > > > > > Leake, David, Zachary Wilkerson, and David Crandall. "Extracting case indices from convolutional neural networks: A comparative study." International Conference on Case-Based Reasoning. Cham: Springer International Publishing, 2022.
> > > > > >
> > > > > > Mertes, Silvan, et al. "Alterfactual Explanations--The Relevance of Irrelevance for Explaining AI Systems." arXiv preprint arXiv:2207.09374 (2022).
> > > > > >
> > > > > > ## Other
> > > > > > Situ, Xuelin. Explaining the Outcomes of Deep Learning Models. Diss. Monash University, 2022.
> > > > > >
> > > > > > Brown, Katherine Elizabeth. Evaluating, Explaining, and Utilizing Model Uncertainty in High-Performing, Opaque Machine Learning Models. Diss. Tennessee Technological University, 2023.
> > > > > >
> > > > > > Jouis, Gaëlle. Explicabilité des modèles profonds et méthodologie pour son évaluation: application aux données textuelles de Pôle emploi. Diss. Nantes Université, 2023.
> > > > > >
> > > > > > Yang, Linyi. Deep Neural Approach for Financial Analysis. Diss. University College Dublin, 2021.
> > > > > >
> > > > > > Owens, Jelani. CLCS: A Convolutional Learning Classifier System for the Surrogate Model in Higher-Dimensional Space. Diss. North Carolina Agricultural and Technical State University, 2022.
> > > > > >
> > > > > > Hong, Xianbin, et al. "Dual-Track Lifelong Machine Learning-Based Fine-Grained Product Quality Analysis." Applied Sciences 13.3 (2023): 1241.
> > > > > >
> > > > > > Buijsman, Stefan. "Defining explanation and explanatory depth in XAI." Minds and Machines 32.3 (2022): 563-584.
> > > > > >
> > > > > > Nimmy, Sonia Farhana, et al. "Interpreting the antecedents of a predicted output by capturing the interdependencies among the system features and their evolution over time." Engineering Applications of Artificial Intelligence 117 (2023): 105596.
> > > > > >
> > > > > > Wen, Bingyang, et al. "Causal-TGAN: Causally-Aware Synthetic Tabular Data Generative Adversarial Network." (2021).
> > > > > >
> > > > > > ## Psychology Literature
> > > > > > Sarasvathy, Saras D. "Even-if: Sufficient, yet unnecessary conditions for worldmaking." Organization Theory 2.2 (2021): 26317877211005785.

---

### Official Review · Reviewer_6hWY · 2023-07-10

**Soundness:** 3 good
**Presentation:** 3 good
**Contribution:** 3 good
**Rating:** 7
**Confidence:** 4

**Summary:**

The paper formalizes the notion of semi-factual for positive outcomes. Given a black-box classifier, a semi-factual is a statement which informs a user about “even if” scenarios in which they would maintain the assigned positive prediction, increasing their “gain”. The authors formalize the problem for both a causal and non-causal setting. They show the validity of their method, S-GEN, via experiments on real-world datasets. Lastly, they also performed a first user study which shows the usefulness of semi-factual for users.

**Strengths:**

To the best of my knowledge, the topic is novel and the proposed ideas are interesting for the algorithmic recourse field. The paper is clear and it can be understood quite well. The related works are discussed extensively and the problem is nicely framed in the context of the state-of-the-art literature. I liked the causal formulation of the semi-factual problem and the user evaluation. More specifically, the user study provides strong hints that semi-factuals have merits, thus it grounds the algorithmic recourse research in the real world.

**Weaknesses:**

The notation and the various proofs can be hard to read at first glance. For example, in Section 3.1., between lines 95 and 112, having $S$ to be the counterfactual and $\textbf{S}$ to be the structural equation instead can cause confusion.

**Questions:**

- If we consider the notion of Gain, how does it relate also to the concept of Shapley value [1]? Which are the differences/similarities between them?
- In Section 3,.4, on line 181, how do you define such distribution density? Namely, given the SCM, you can define the interventional joint distribution after a do action $Pr(\mathbf{X} | do(\mathbf{a}))$. Is it the same thing?
- In Section 4, how do you define the “empirical approximation” of $G$, $P$, $H$, $R$ and $J$? How can you learn such approximations?
- In Section 6, on line 333, the paper says “...rate on a scale from 1-5 how useful each where.”. Could you elaborate more on this point? Useful to what?

[1] Lundberg, Scott M., and Su-In Lee. "A unified approach to interpreting model predictions." Advances in neural information processing systems 30 (2017).


**Limitations:**

The authors adequately address all the limitations.

---

> ### Author Rebuttal · Authors · 2023-08-07
>
> We greatly thank the reviewer for the extremely positive comments, noting the novelty of our work, and voting for acceptance. We now address all questions and concerns:
>
> ***
>
> *The notation and the various proofs can be hard to read at first glance. For example, in Section 3.1., between lines 95 and 112, having $S$ to be the counterfactual and $\mathbf{S}$ to be the structural equation instead can cause confusion.*
>
> Thanks for making us aware of this, we have changed $S$ to $SF$ for representing the semi-factual function to avoid confusion here. If there is anything else specific which stands out we can modify that also, please just let us know.
>
> **Revision:** We modified the notation for $S$ => $SF$ in Section 3.1
>
> ***
>
> *If we consider the notion of Gain, how does it relate also to the concept of Shapley value [1]? Which are the differences/similarities between them?*
>
> In XAI, Shapley values are usually used to explain the contribution of each feature to a prediction (or score) either for a single sample or a set of samples. Gain is a function which outputs a real-valued number representing how much a user benefited (positively or negatively) from feature mutations. We feel the two have few if any similarities.
>
> ***
>
> *In Section 3,.4, on line 181, how do you define such distribution density? Namely, given the SCM, you can define the interventional joint distribution after a do action $Pr(\mathbf{X}| do(\mathbf{a}))$. Is it the same thing?*
>
> Yes you are correct it is the same thing. Assume that we read $Pr(\mathbf{X}| do(\mathbf{a}))$ as the probability of $\mathbf{x}$ resulting from some $\mathbf{x_0}$ taking action $\mathbf{a}$. $Pr(S_{M}(\mathbf{x}, \mathbf{a}))$ is the probability of some $\mathbf{x}’$ resulting from $\mathbf{x}$ taking action $\mathbf{a}$. Essentially, $\mathbf{x_0}$ and $\mathbf{x}$ are the same as $\mathbf{x}$ and $\mathbf{x’}$ in the notation, respectively. If the reviewer prefers we can easily change the notation to match what you wrote above, or simply note this in a footnote, please let us know your preference.
>
> **Revision:** We await your feedback.
>
> ***
>
> *In Section 4, how do you define the “empirical approximation” of G, P, H, R and J? How can you learn such approximations?*
>
> We define the empirical approximations as follows: $G$ is a simple euclidean distance between a test instance and the generated semi-factual(s) (see lines 294-297). $P$ is defined (in non-causal tests) in line 257, it is a simple function to return high values when the test instance is close to a training datapoint (high plausibility), and low values when it’s not (low plausibility), it is naturally taken care of in causal domains thanks to the SCM. $H$ is the Lagrangian interpretation of robustness $R$, it is approximated with $\hat{H}_p$ which uses MC sampling to see if the neighborhood around the semi-factual explanation(s) is robust, and $\hat{H}_a$ which checks the robustness of the semi-factual(s) itself by classifying it with the model (i.e., it doesn’t need to be approximated). $J$ is just the objective.
>
> There is a vast literature on learning optimal weights in multi-objective optimization [4], we chose one of the most popular approaches of a grid search (see line 237), but other options involving reinforcement learning, bayesian optimization etc. are options. But in our case it wasn’t necessary to achieve the desired performance.
>
> ***
>
> *In Section 6, on line 333, the paper says “...rate on a scale from 1-5 how useful each where.”. Could you elaborate more on this point? Useful to what?*
>
> No problem, there are many things the semi-factuals could be useful for in the context of a loan application. Our take is that they are useful for things such as e.g. spending more time with family (see lines 131-133), or reducing your downpayment to have more flexible financial options. We didn’t want to explicitly define what was meant by “useful” to users (indeed we avoided it in the user study on purpose), but rather let people use their own “natural” interpretation. This is standard practice in cognitive studies in our experience, and indeed the same approach is taken in the DARPA “trust” and “satisfaction” measures too which are popular in XAI [2]. A sign this has failed is if the results are not consistent (e.g. see [3] and their "reasonableness" measure), but here we see a clear result in that people seem to converge on what they interpret as “useful” for the setting.
>
> **Revision:** We explain our definition of “useful” in Section 6.
>
> ***
>
> [1] Lundberg, Scott M., and Su-In Lee. "A unified approach to interpreting model predictions." Advances in neural information processing systems 30 (2017).
>
> [2] Hoffman, Robert R., et al. "Metrics for explainable AI: Challenges and prospects." arXiv preprint arXiv:1812.04608 (2018).
>
> [3] Kenny, Eoin M., et al. "Explaining black-box classifiers using post-hoc explanations-by-example: The effect of explanations and error-rates in XAI user studies." Artificial Intelligence 294 (2021): 103459.
>
> [4] Gennert, Michael A., and Alan L. Yuille. "Determining the optimal weights in multiple objective function optimization." ICCV. 1988

---

> > ### Comment · Reviewer_6hWY · 2023-08-11
> >
> > I thank the authors for the extensive rebuttal and for clearing my doubts! I do not have additional questions since I think the suggestions and improvements the authors made (also for the other reviewers) resolved most of the issues.

---

### Official Review · Reviewer_LWa1 · 2023-07-15

**Soundness:** 3 good
**Presentation:** 1 poor
**Contribution:** 3 good
**Rating:** 6
**Confidence:** 2

**Summary:**

This paper explores techniques for optimizing positive outcomes using semi-factuals. To do so, they propose the concept of "gain" and set up a small user test to demonstrate the usefulness of their technique.

**Strengths:**

The authors offer an interesting and compelling framework for optimizing positive outcomes, with strong formal notation and proofs.

**Weaknesses:**

My biggest concern with the paper is its presentation. I think the authors still have much work to do in cleaning up the writing and presentation of this work. See below.

**Questions:**

-- I think most importantly, the authors need to better motivate their work, especially in the Introduction. I think it might help to give more context of previous work in the Introduction: in what contexts is this kind of work used? what's come before? how does this work add on to it? The answers to these questions are scattered throughout the rest of the paper, and in Related Work, but it should be very crisply and tightly explained in the Introduction. In general, the language in the Introduction is a little too loose. I'd love a very clear distinction between counterfactuals and semi-factuals, and between positive outcomes and negative outcomes. I see that loans are one context in which this work could be useful, but what are other contexts in which this work could be useful? This is just a smattering of ideas for cleaning up the Introduction, but I think in general, the Introduction feels a little too philosophical. At some point in the Introduction, I'd like it to be made clear how this is used in the context of ML.
-- There are also some strong claims made that I would want justified (e.g. "completely unexplored" in line 22).
-- I wasn't convinced by the two main issues outlined in the second paragraph of the Introduction. These are the two main issues according to who? Fairness is mentioned in line 37, but it wasn't clear how fairness and ethics is really an issue in this paragraph. (I understand what the authors are trying to say, but the writing doesn't make it as clear as it could be.)
-- There are typos scattered throughout the manuscript, which should be proof-read carefully.
-- It might help to move part of lines 61-65 into the Introduction. Also, I think describing the "urgent need" in line 57 would help motivate this work, too.

**Limitations:**

The authors do mention the limitations of their user evaluation in the Discussion, but I feel they could expand on the limitations of their approach a bit further.

---

> ### Author Rebuttal · Authors · 2023-08-07
>
> We thank the reviewer for noting the compelling nature of our framework, and on account of voting for the acceptance of the work. Here, we address your specific questions and concerns.
>
> ***
>
> *I think... the authors need to better motivate their work, especially in the Introduction... the language... is a little too loose. I'd love a very clear distinction between cfs and sfs, and between positive v. negative outcomes.*
>
> Thank you for pointing this out. We have added clearer definitions for CFs v. SFs and positive v. negative outcomes to the intro. the new sentences added state: *“Our definition of counterfactuals is in line with the literature and Wachter et al. [4], where a given test instance $x$ classified as class $c$ must be modified in such a way as to cross a decision boundary into class $c'$, and a semi-factual $x$ classified as $c$ must be modified in such a way as to not cross a decision boundary (and hence remain class $c$) [5]”*, and *“In recourse, bad outcomes (e.g., a loan rejection) are generally mutated to produce good outcomes (e.g., a loan acceptance) for users using counterfactuals. In our setting, we are assuming there was initially a good outcome, and we are trying to mutate features to produce an even better outcome for users, but in doing so not cross the boundary into the bad outcome (i.e., using semi-factuals).”*
>
> **Revision:** We added the above sentences to the intro and “tightened up” the language to be less loose in general and more formal.
>
> ***
>
> *Where else is this work useful?*
>
> Good question, we envision that it will also be quite useful in medical applications (partly why we used the BCSC dataset). For example, patients are often over prescribed drugs with adverse side effects [1]. So we envision a semi-factual could say *“Even if you half your intake of drug x, you will still be at a low risk for disease y”*. This way the patient can hopefully avoid the side effects of taking so much of the drug, whilst still maintaining low disease risk (i.e., the good outcome). But the explanations will be likely useful in any application that can be parsed as positive/negative outcomes.
>
> **Revision:** We note the possible medical applications for our work in the second paragraph of Sec 1.
> ***
> *In the Introduction, I'd like it to be made clear how this is used in the context of ML.*
>
> We understand, we have mentioned how we envision the explanations being useful for financial and medical applications as discussed above, as well as a clear definition of CF v. SF and positive v. negative outcomes. Apologies for the lack of clarity.
>
> **Revision:** See above two revisions.
> ***
> *Justify "completely unexplored" in line 22.*
>
> Noted, we conducted a literature review to the best of our ability and couldn’t find comparable work.
>
> **Revision:** We have rephrased this to *“... which (to the best of our knowledge) remains completely unexplored”* to be more conservative.
> ***
> *I wasn't convinced by the two main issues outlined in the second paragraph of the Introduction. These are the two main issues according to who?*
>
> Artelt & Hammer [2]  discuss the second issue as a concern (i.e., manipulating people to accepting bad outcomes when they don’t have to) which we cite (line 22). Now, regarding the first issue of “usefulness”, we found this through our user study both through (1) the results, and (2) initial pilot testing to gauge people’s feelings about the explanations. However there are other possible issues such as how they can negatively influence people’s causal understanding of AI systems [3], but this was beyond our scope to investigate here.
>
> **Revision:** We have fixed this by changing line 26 to:  *“However, such a use case for semi-factual explanation **perhaps** has two main issues”*. Also, we foreshadow the user study results to make this statement about “usefulness” more convincing.
>
> ***
>
> *...it wasn't clear how fairness and ethics is really an issue… It might help to move part of lines 61-65 into the Introduction.*
>
> Thank you for this. We have done as you suggested and we agree it does indeed make the fairness issues much clearer in the introduction, thanks for the suggestion.
>
> **Revision:** We moved lines 61-65 to Paragraph 2 in the Intro.
>
> ***
>
> *There are typos scattered throughout the manuscript, which should be proof-read carefully.*
>
> Apologies for those mistakes, fixed.
>
> ***
>
> *Also, I think describing the "urgent need" in line 57 would help motivate this work, too.*
>
> Good point, we have elaborated with the recently published work [3] as motivation here as they discuss *“the paucity of user studies”* as a severe limitation for semi-factuals; in that until some carefully-controlled studies are carried out, we do not really know how users will respond to these explanations in the ML context.
>
> ***
>
> *The authors do mention the limitations of their user evaluation in the Discussion, but I feel they could expand on the limitations of their approach a bit further.*
>
> Agreed, we added the limitation regarding the need for a causal model of the domain in order to use causality when calculating gain (although you may still do this without the SCM in our non-causal algorithm).
>
> ***
>
> [1] Safer, Daniel J. "Overprescribed medications for US adults: four major examples." Journal of clinical medicine research 11.9 (2019): 617.
>
> [2] Artelt, André, and Barbara Hammer. "“Even if…”–Diverse Semifactual Explanations of Reject." 2022 IEEE Symposium Series on Computational Intelligence (SSCI). IEEE, 2022.
>
> [3] Aryal, Saugat, and Mark T. Keane. "Even if explanations: Prior work, desiderata & benchmarks for semi-factual XAI." IJCAI Survey Track – 2023
>
> [4] Wachter, Sandra, Brent Mittelstadt, and Chris Russell. "Counterfactual explanations without opening the black box: Automated decisions and the GDPR." Harv. JL & Tech. 31 (2017): 841.
>
> [5] Kenny, Eoin M., and Mark T. Keane. "On generating plausible counterfactual and semi-factual explanations for deep learning." AAAI

---

> > ### Comment · Reviewer_LWa1 · 2023-08-14
> >
> > I thank the authors for their reply! I appreciate the updates. I think it's hard to evaluate the clarity of the new draft without being able to see it, but assuming the clarifications are adequate, I'll change my score to a 6.

---

> > > ### Author Response · Authors · 2023-08-17
> > > **Author(s) Followup**
> > >
> > > We thank the reviewer for their followup and engagement with our work!
> > >
> > > We noticed the reviewer didn’t actually update their score, and we are just wondering if this was a simple oversight, or if you prefer to see larger portions of the new draft to do so? We note the primary areas of concern are simply (1) the Lit. Review paragraph about user studies, (2) the limitations, and (3) the intro.
> > >
> > > If so, we are happy to present the relevant revised paragraphs here in a comment, please just let us know.

---

> > > > ### Comment · Reviewer_LWa1 · 2023-08-17
> > > >
> > > > This was an oversight—thank you for catching that!

---

### Author Rebuttal · Authors · 2023-08-07

We would like to thank the reviewers for taking the time to read our paper and give constructive feedback. We have taken seriously the suggestions of all reviewers and either amended the manuscript accordingly to address the concerns, or discussed it here in our response.

Firstly however, we thank the reviewers for their positive comments, and overall vouching for acceptance, **Reviewer-6hWY** said *“To the best of my knowledge, the topic is novel and the proposed ideas are interesting for the algorithmic recourse field. The paper is clear and it can be understood quite well. The related works are discussed extensively and the problem is nicely framed in the context of the state-of-the-art literature. I liked the causal formulation of the semi-factual problem and the user evaluation. More specifically, the user study provides strong hints that semi-factuals have merits, thus it grounds the algorithmic recourse research in the real world.”* **Reviewer-vFfC** pointed out that *“the paper tackles a new problem that has not had much attention in the XAI community – Another plus is the simplicity of the idea which i like very much given that it is well-motivated”*, **Reviewer-LWa1** stated *“The authors offer an interesting and compelling framework for optimizing positive outcomes, with strong formal notation and proofs.”* lastly **Reviewer-4usU** said *“the human study at the end is very cool - I haven't seen so many of these in this literature and I think that really sets this paper apart”*.

We are truly grateful for these (and other) encouraging comments made.

Beyond the positive reviews, we are grateful for many of the suggestions provided (mainly presentation related) and have incorporated them into the latest revision to improve the manuscript. Our changes include:

* We took the suggestions of **Reviewer LWa1** and improved the clarity of our writing in the introduction, literature review, and limitations (details in our response to the reviewer).
* Regarding the feedback of **Reviewer 6hWY**, we modified the notation of our semi-factual function and also made our notion of what “useful” means clear in the user study.
* Thanks to **Reviewer vFfC** we make it clearer why gain and cost are not the same and how SCMs guarantee our notion of plausibility.
* **Reviewer 4usU** overall felt the paper lacked clarity in pages 3-6, so we added additional detail around areas of confusion alongside other suggestions from the reviewer to improve the presentation (details in response to the reviewer).
* For **Reviewer 4usU** we also uploaded the non-normalized results of Experiment 1 for clarity here as a PDF file (our non-causal tests), we will add this to the final appendices in case readers are interested.

**Importantly, all of these were quite minor edits that we completed the past week**. Although we are not allowed to upload the updated manuscript, we detail as much as appropriate the changes made below.

Best wishes,

Anonymous Author(s)

---

### Decision · Program_Chairs · 2023-09-21

**Decision:**

Accept (poster)

**Comment:**

The authors formalizes the notion of semi-factual for positive outcomes and propose a methodology to optimize them. A majority of reviewers were positive about the paper. There was one reviewer who was particularly negative about the paper, however, the reviewer's concerns were mainly due to a lack of clarity in terms of presentation, which was also shared by other reviewers. I request the authors to revise their manuscript very carefully following the reviews/rebuttal when preparing the final version of the paper.